# An enzymatic continuous-flow reactor based on a pore-size matching nano- and isoporous block copolymer membrane

Zhenzhen Zhang [1,5], Liang Gao [2,5], Alexander Boes [3], Barbara Bajer[1], Johanna Stotz [2], Lina Apitius[3], Felix Jakob [3], Erik S. Schneider[1], Evgeni Sperling[1], Martin Held [1], Thomas Emmler [1], Ulrich Schwaneberg [2,3] ✉ & Volker Abetz [1,4] ✉

Continuous-flow biocatalysis utilizing immobilized enzymes emerged as a sustainable route for chemical synthesis. However, inadequate biocatalytic efficiency from current flow reactors, caused by non-productive enzyme immobilization or enzyme-carrier mismatches in size, hampers its widespread application. Here, we demonstrate a general-applicable and robust approach for the fabrication of a high-performance enzymatic continuous-flow reactor via integrating well-designed scalable isoporous block copolymer (BCP) membranes as carriers with an oriented and productive immobilization employing material binding peptides (MBP). Densely packed uniform enzyme-matched nanochannels of well-designed BCP membranes endow the desired nanoconfined environments towards a productive immobilized phytase. Tuning nanochannel properties can further regulate the complex reaction process and fortify the catalytic performance. The synergistic design of enzyme-matched carriers and efficient enzyme immobilization empowers an excellent catalytic performance with >1 month operational stability, superior productivity, and a high space-time yield ($1.05 \times 10^5 \, g \, L^{-1} \, d^{-1}$) via a single-pass continuous-flow process. The obtained performance makes the designed nano- and isoporous block copolymer membrane reactor highly attractive for industrial applications.

Biocatalysis with immobilized enzymes applied in continuous flow has emerged as an appealing and sustainable approach to biocatalytically produce chemicals through increased reaction productivity, improved reaction control, scale-up efficiency, and downstream processing[1]. Although the immobilization of enzymes within solid carriers facilitates enzyme/product separation, it usually hinders the mass transfer of substrates toward the enzyme's active site[2]. The latter can often be attributed to differences in enzyme sizes (often 4–12 nm) when compared with pores/cavities in carrier systems that are usually orders of magnitudes larger. Recently developed continuous-flow reactors have been equipped with a miniaturized carrier space to tackle this obstacle via reducing the mass transfer distance and enriching the local enzyme concentration[3–5], including column reactors with micro/nanoparticles[6] or microcapsules[7–9], microfluidic reactors[10], polymeric/anodic aluminum oxide (AAO) membrane reactors[11,12]. However, these systems still encounter at least one of the following issues: (i) enzyme-

[1]Helmholtz-Zentrum Hereon, Institute of Membrane Research, Max-Planck-Straße 1, 21502 Geesthacht, Germany. [2]RWTH Aachen University, Institute of Biotechnology, Worringerweg 3, 52074 Aachen, Germany. [3]DWI—Leibniz-Institute for Interactive Materials, Forckenbeckstraße 50, 52056 Aachen, Germany. [4]Universität Hamburg, Institute of Physical Chemistry, Martin-Luther-King-Platz 6, 20146 Hamburg, Germany. [5]These authors contributed equally: Zhenzhen Zhang, Liang Gao. ✉e-mail: u.schwaneberg@biotec.rwth-aachen.de; volker.abetz@hereon.de

mismatched carrier spaces/nanochannels, which may be either too narrow for enzyme entry/immobilization or too wide for efficient substrate-to-product conversion at high mass transfers; (ii) uneven size distribution of carrier nanochannels, which may lead to an insufficient catalytic performance under uneven enzyme distributions and uncontrollable reaction conditions; (iii) to a less extend non-productive enzyme immobilizations or leaching via physisorption and/or chemical modifications. Here, to address all these issues, we provide a general-applicable and robust enzymatic flow reactor design system through the combination of well-designed scalable isoporous block copolymer (BCP) membranes as carriers with an oriented one-step enzyme immobilization via a genetically fused material binding peptide (MBP). MBPs enable an enzyme immobilization with high surface coverages of >80% without affecting the enzyme flexibility/activity at ambient temperature from aqueous solutions[13].

Isoporous BCP membranes possess well-ordered, uniform, enzyme-matched nanosized channels (10–100 nm), a high pore number density (>$10^{14}$ pores m$^{-2}$), thereby as well as a high surface area-to-volume ratio[14–16]. Therefore, BCP membranes not only enable as carriers an efficient protein immobilization/enrichment with sufficient protein surface coverage[17], but also ensure efficient mass transfer of substrates to the active site of enzymes[4,5] and offers precise reaction control[18,19]. BCP membranes are extensively exploited as advanced membrane separation materials[20–23], thus enabling the integration of catalytic reaction and product separation in one unit and offering a scalable and efficient process intensification. In this work, BCP membranes were fabricated via the combination of evaporation-induced self-assembly and nonsolvent-induced phase separation (SNIPS)[24]. The latter one-step approach is straightforward and scalable to produce asymmetric but integral membranes, which exhibit a unique structure with an isoporous top layer and a macroporous supporting sublayer[14]. These features endow BCP membranes with superior mechanical robustness and scalability over the conventional AAO isoporous membranes which are thermally stable and produced in a multiple-step process[12]. Compared to track-etched isoporous membranes[25], BCP membranes possess higher porosity with 2 orders of magnitudes higher pore number density and thereby superior permeability. Additionally, tailoring the nanochannel properties of carriers confers the on-demand regulation on the reaction process[6,26]. For SNIPS BCP membranes, the nanochannel size can be tuned down to sub-10 nm and the nanochannel surface properties can be adjusted from hydrophilic to hydrophobic and from neutral to charged[20,27–29]. Thus, SNIPS BCP membranes can be systematically varied in a wide range, which makes them highly adaptive to overcome unspecific enzyme immobilization and reaction process challenges.

MBPs (adhesion-promoting peptides or anchor peptides) can strongly bind to diverse natural and synthetic material surfaces with high affinity and efficiency relying on noncovalent multiple-site interactions (e.g., electrostatic, hydrophobic, π-π interactions, and hydrogen bonds)[30–32]. Genetic fusion of MBPs with various biomolecules (e.g., enzymes, proteins) results in MBP-enzyme fusions that can be efficiently immobilized in an oriented manner onto carriers[13,33,34]. Favourably oriented immobilization of enzymes and a spacer helix between the MBP and enzymes ensure a special separation of the binding and catalytic enzyme part, and thereby an unaltered catalytic enzyme property[13,35]. Moreover, binding strength and material-specific binding of MBPs can be adjusted with standard protein engineering methodologies to match requirements of application conditions (e.g., salt, pH, temperature, product/substrate, or detergent concentration)[30,36–38].

Phytases are industrially important enzymes used in animal nutrition[39]. Phytases catalyze a stepwise release of phosphate from the phosphate storage compound phytate of plants[40]. Recovery of phosphate from phytate in rapeseed and other plants offers a route to green phosphate that could cover 30% of the phosphate fertilizer

consumption in Germany and valorisation opportunities to flame retardant polyphosphate compounds[41,42]. Efficient phytate hydrolysing enzymatic flow reactors are the prerequisite for a circular P-economy and higher value-added products like polyphosphate.

In this work, we prepare the enzymatic continuous-flow reactor using the polystyrene-*block*-poly(4-vinyl pyridine) (PS-*b*-P4VP) isoporous membranes together with the MBP named Liquid Chromatography peak I (LCI) fused to a phytase from *Yersinia mollaretii* (YmPh). Our reactor system enables efficient catalysis within milliseconds in a continuous single-pass process with superior operational stability (> one month) and high productivity (Fig. 1). Building on the design flexibility of BCP membranes as well as the scalable and cost-effective MBP immobilization, our reactor system illustrates its biocatalytic potential. The enzyme-functionalized nano- and isoporous block copolymer membrane reactor operates at continuous flow with high-performance, which we term as "nanoporous membrane reactor" (NaMeR) in the following text.

## Results

### Phytase immobilization on nano- and isoporous BCP membrane

In our design, the industrially important phytase YmPh, the MBP LCI, and a polystyrene-*block*-poly(4-vinyl pyridine) (PS-*b*-P4VP) BCP membrane were combined to produce phosphate from phytate. To investigate the efficiency of YmPh immobilization to the PS-*b*-P4VP isoporous membrane, wild-type YmPh (YmPh-WT as reference) and LCI-fused YmPh (YmPh-LCI, Supplementary Notes 4.1–4.3) were prepared and compared[32]. YmPh-WT and YmPh-LCI possess a comparable hydrodynamic diameter of approx. 6.2 nm and a comparable enzymatic activity (Fig. 2a, Supplementary Figs. 1–3). Meanwhile, the designed BCP membrane, having an isoporous cylindrical top layer with a thickness of approx. 350 nm supported by a macroporous spongy sublayer of approx. 40 μm thickness (Fig. 2b), was fabricated from the well-defined PS-*b*-P4VP diblock copolymer (Supplementary Table 1)[43]. The densely packed, uniform nanochannels have a diameter of 57.5 ± 1.9 nm in the cylindrical top layer. The diameter is only ~9-times the size of the YmPh-LCI, which provides a nanoconfined environment for the enzymatic phytate hydrolysis reaction, while there is still a sufficient radial space for the enzyme binding along the nanochannel wall (final nanochannel diameter of approx. 45.1 nm, assuming uniform coating)[44].

After optimizing immobilization conditions (Supplementary Figs. 4,5), the YmPh-LCI immobilized on the membrane (YmPh-LCI@M, @M indicates membrane-bound YmPh) displayed an approx. three orders of magnitudes higher activity and binding capacity (830 pmol cm$^{-2}$) when compared to the wild-type YmPh-WT@M reference (Fig. 2c, Supplementary Table 2). Significantly improved catalytic activity of YmPh-LCI over YmPh-WT can directly be attributed to its efficient immobilization through the adhesion-promoting LCI peptide. YmPh-LCI onto a PS-*b*-P4VP substrate was further characterized by atomic force microscopy (AFM). YmPh-LCI formed a homogeneous layer with a thickness of approx. 7.0 nm onto the dense PS-*b*-P4VP film (Fig. 2d,e, Supplementary Figs. 6,7, Supplementary Table 3), which is in good agreement with the calculated size and the measured hydrodynamic diameter of YmPh-LCI (Fig. 1, Supplementary Fig. 1b). It indicates that the bound YmPh-LCI forms a monolayer without layer-by-layer adsorptions or aggregates, which is beneficial to retain the native conformation and activity of the phytase.

To further probe the location of YmPh-LCI@M within the asymmetric membrane nanostructure, YmPh-LCI was labelled with a fluorescent dye (sulfo-cyanine3 NHS ester) as well as with gold nanoparticles (1.4 nm Mono-Sulfo-NHS-Nanogold®) (Supplementary Figs. 8,9); the distribution of fluorescence and gold nanoparticles along the cross-section of the membrane was determined respectively. Fluorescence is intense in an approx. 1 μm top layer of the membrane (Fig. 2f, Supplementary Figs. 10,11), while the gold signal is stronger in

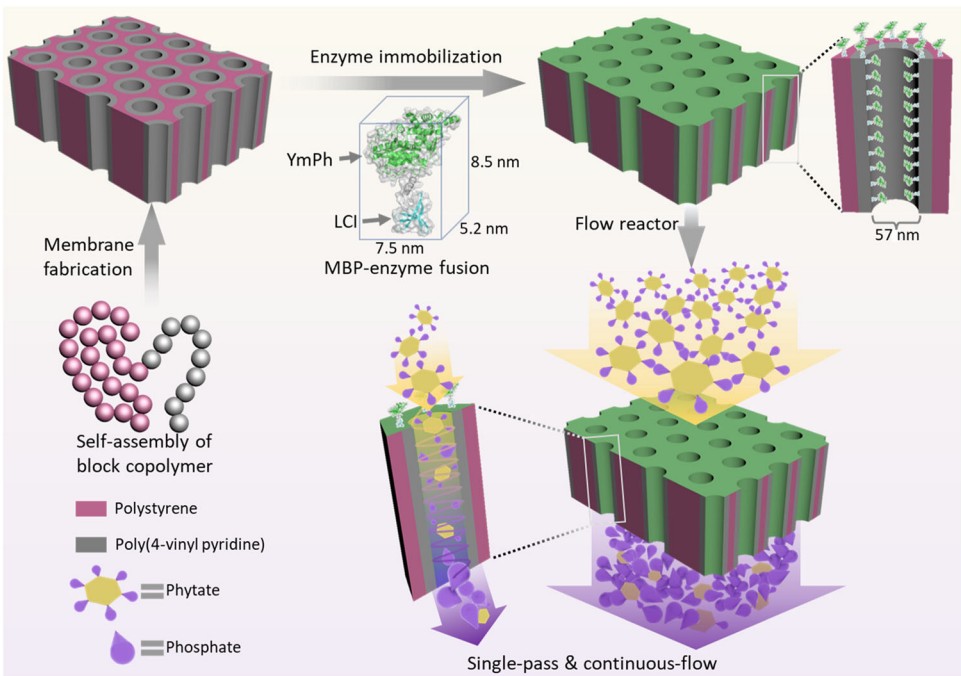

**Fig. 1 | Schematic illustration of the general-applicable and scalable flow reactor design by combining well-designed isoporous BCP membranes as carriers with a fusion protein (phytase plus material binding peptide (MBP; namely LCI)) for oriented phytase (YmPh) immobilization.** High performance of the designed NaMeR is exemplified by matching the size of the membrane nanochannel and the YmPh-LCI fusion protein. Efficient stepwise phosphate release from phytate is achieved when the phytate feed solution passes through the nanochannels with immobilized YmPh-LCI in a single-pass continuous flow. Parts of the diagram were adapted from ref. 20 (CC BY 4.0 DEED) and ref. 29 (CC BY 3.0 DEED).

the top layer up to approx. 370 nm, matching the thickness of the isoporous cylindrical top layer (Fig. 2b, g). Furthermore, the gold-labelled YmPh-LCI was detected as tiny dark particles accumulating on the nanochannel surface of the cylindrical top layer (Fig. 2h, i). Therefore, it is clear that YmPh-LCI@M predominantly localizes within the isoporous cylindrical top layer rather than within the macroporous spongy sublayer, which is mainly attributed to the much higher surface–to-volume ratio as well as the designed enzyme-matched nanochannel size in the cylindrical top layer[17]. Overall, the oriented, homogeneous, and material-specific YmPh-LCI immobilization in the designed isoporous cylindrical top layer paves the way towards a general development of isoporous enzymatic continuous-flow membrane reactors, in which nanochannel and enzyme sizes match to efficiently convert small substrate molecules.

## Performance of the continuous-flow nano- and isoporous membrane reactor

We performed a pressure-driven continuous-flow reaction via a single-pass process in dead-end mode (flow-through configuration, Fig. 3a, Supplementary Fig. 12) and determined the stepwise phosphate ($PO_4^{3-}$) release from phytate (myo-inositol-1,2,3,4,5,6-hexakisphophate or InsP6) employing YmPh-LCI. The stepwise hydrolysis can be considered as a cascade flow reaction, in which the substrate (InsP6) and intermediates (lower phosphorylated myo-inositol, InsP) 'collide' and are hydrolysed by several phytase enzymes along their way through the nanochannel. We assessed the phytate hydrolysis performance by determining the phosphate concentrations ($C_{PO_4^{3-}}$) in the permeate (Supplementary Fig. 13), the average amount of hydrolysed $PO_4^{3-}$ per InsP6 (denoted as $R_P$), and productivity (defined as the amount of produced phosphate from hydrolysed InsP6 per membrane reactor area per unit time). Additionally, different phosphatidylinositol intermediates from phytate (InsP6) are generated in the stepwise phosphate hydrolysis reactions ranging from InsP5, InsP4, InsP3, InsP2, InsP1 to finally inositol and can be composed of varied isomer

compositions depending on the phosphate cleavage preferences[39,45]. Hence, the maximum catalytic performance (i.e., $C_{PO_4^{3-}}$ and $R_P$) of free YmPh-LCI was determined in solution with varied phytate concentrations as a reference. Key process parameters of the continuous-flow membrane reactor, including flux and InsP6 concentration ($C_{InsP6}$) that directly determine how fast or how many molecules through flow reactors, were investigated to determine its performance.

Flux determines the mass transfer rate of substrate through the nanochannels and the residence time of substrate inside the membrane nanochannels[46,47]. With increasing flux from 6 to 215 L m$^{-2}$ h$^{-1}$, the residence time of InsP6 inside the membrane nanochannels varies from $7.0 \times 10^{-2}$ to $0.2 \times 10^{-2}$ s, which is much longer than the radial molecular diffusion time ($-2.5 \times 10^{-6}$ s; a detailed calculation is provided in the Supplementary Figs. 14,15, Supplementary Tables 4,5). Sufficient residence time (4 orders of magnitudes higher than molecular diffusion time) proves that the designed nanochannels enable the substrate stream to intensively interact and collide with the nanochannel wall without a radial molecular diffusion limitation. Therefore, matching the enzyme and nanochannel size demands is the prerequisite for a high-performance continuous-flow reactor (Fig. 3a). Additionally, within a low flux range (6–15 L m$^{-2}$ h$^{-1}$), the YmPh-LCI@M performance shows a maximal $C_{PO_4^{3-}}$ of 1.89 mM, a $R_P$ of 4.9, and a phosphate release efficiency of 83%, comparable to the maximum catalytic performance of non-immobilized/free YmPh-LCI (Fig. 3b, Supplementary Fig. 16). A further increase of flux to 215 L m$^{-2}$ h$^{-1}$ improves the productivity sharply to the maximal value of 272 mmol m$^{-2}$ h$^{-1}$ due to an increased mass transfer rate (bringing more substrates into nanochannels per unit time) that may enhance specific phytase activity, while it decreases $C_{PO_4^{3-}}$ to 1.26 mM, $R_P$ to 3.3, and the phosphate release efficiency to 55% (Fig. 3b, Supplementary Table 6). Such decline arises from the reduced contact time between the substrate and immobilized enzyme rather than the YmPh-LCI detachment, since the catalytic performance is almost completely recovered after a high pressure-driven flow of 215 L m$^{-2}$ h$^{-1}$ (star point at Fig. 3b)[47]. For

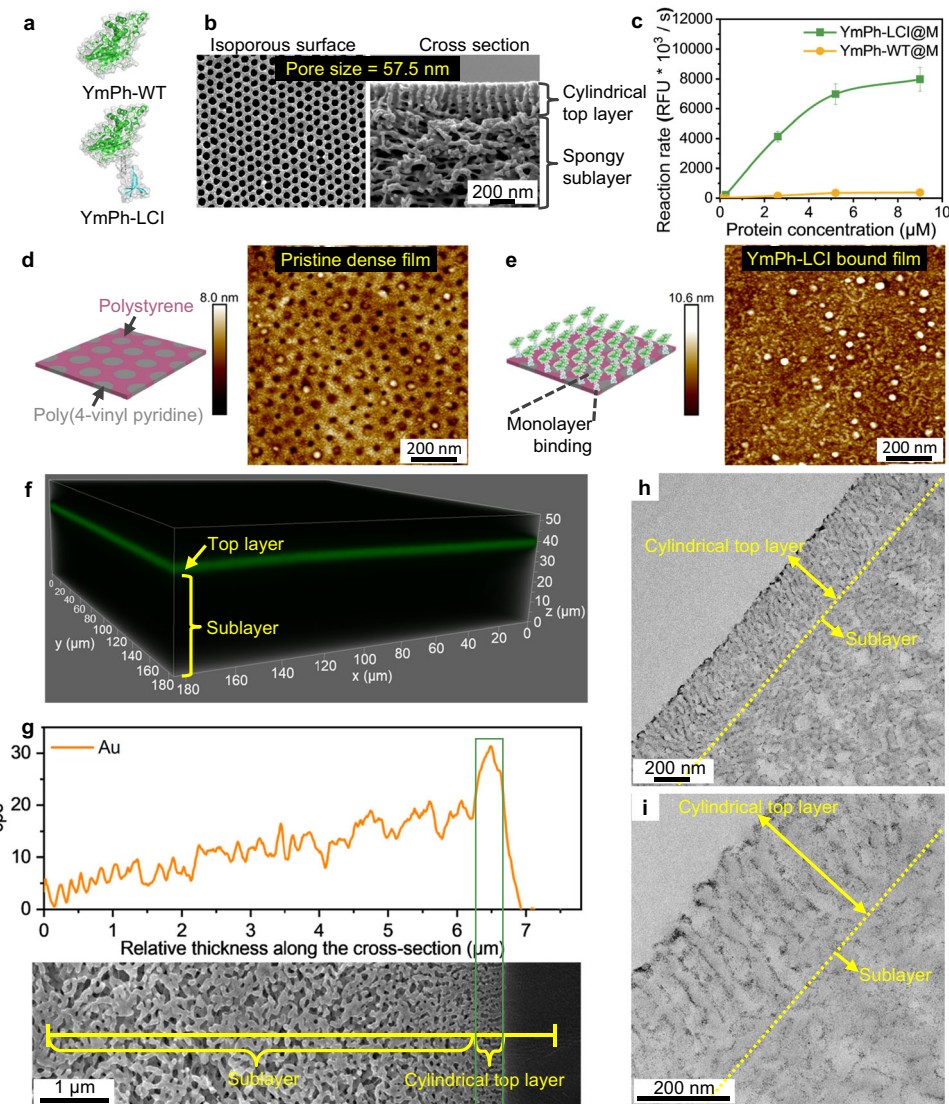

**Fig. 2 | Enzyme immobilization on nano- and isoporous BCP membranes. a** 3D structure of wild-type phytase (YmPh-WT) and YmPh genetically fused to the MBP LCI (YmPh-LCI). **b** SEM images of PS-*b*-P4VP isoporous membranes: top surface and cross-section. **c** Dose-response curve to analyze binding efficiency of YmPh-WT and YmPh-LCI on isoporous BCP membranes (@M indicates membrane-bound YmPh). Error bars represent s.d. of the mean from three independent experiments (*n* = 3). AFM height images of dense PS-*b*-P4VP films (**d**) treated with buffer solution as a control (Tris-HCl buffer (50 mM, pH 8.0)) and (**e**) after YmPh-LCI immobilization. **f** 3D reconstruction image from a z-stack along the cross-section of the membranes

with fluorescently labelled YmPh-LCI_Cy3. The membrane's top layer faces up. **g** A reconstructed elemental line scan of the gold signal along the cross-section of the membranes with gold-labelled YmPh-LCI_Au from energy-dispersive X-ray spectroscopy (EDX) mapping. The membrane's top layer faces right. Transmission electron microscopy (TEM) bright-field images of the cross-section of the membranes with YmPh-LCI_Au: (**h**) low magnification, (**i**) high magnification. Results were reproduced three times independently; representative micrographs are shown. Source data are provided as a Source Data file.

phytases from *A. niger*, *A. terreus*, and *E. coli*, a significant decrease in hydrolysis rates during phytate degradation was reported, leading to the accumulation of lower InsP intermediates (e.g., ≤InsP4)[45]. Our designed continuous-flow NaMeR enables multi-step hydrolysis of InsP6 to InsP3 within milliseconds at a high flux of 215 L m$^{-2}$ h$^{-1}$ and confers a superior efficiency (Fig. 3b, Supplementary Fig. 16, Supplementary Table 6). This arises from the synergistic effect of the enriched local enzyme surface concentration and the minimized enzyme-enzyme/enzyme-substrate distance in the designed enzyme-matched nanochannels (i.e., nanoconfinement effect[48]). Such synergistic effect enables a rapid conversion of InsP6 and generates a stepwise high local concentration gradient of converted InsP intermediates within the nanochannel; the latter ensures a high specific enzyme activity for rapid hydrolysis of InsP6 to InsP3. The achieved multistep hydrolysis proves that the continuous-flow NaMeR can be used for single enzyme

conversions ranging from one-step to multistep reactions, and can therefore in principle be employed for multienzyme cascade reactions if enzymes could be placed in a defined manner within membrane nanochannels. Based on the obtained results, we recommend to use the continuous-flow NaMeR either with a high flux of 215 L m$^{-2}$ h$^{-1}$ to maximize productivity or with a low flux (e.g., ≤15 L m$^{-2}$ h$^{-1}$) to maximize phosphate yields from one phytate molecule.

An increased substrate concentration will force more substrates to contact the enzyme-coated nanochannels, leading to stepwise hydrolysis of InsP6 to InsP Intermediates (Fig. 3a). With increasing $C_{InsP6}$ up to 23.15 mM, YmPh-LCI@M displays a nearly linear increase of $C_{PO_4^{3-}}$ to 93.81 mM, a steady $R_P$ (4.0–4.3), and a steady phosphate release efficiency (67–72%), regardless of a sharp decline of $R_P$ in the low $C_{InsP6}$ range (<0.8 mM) (Fig. 3c, Supplementary Tables 7,8). Notably, this trend is similar to that of the maximum catalytic performance

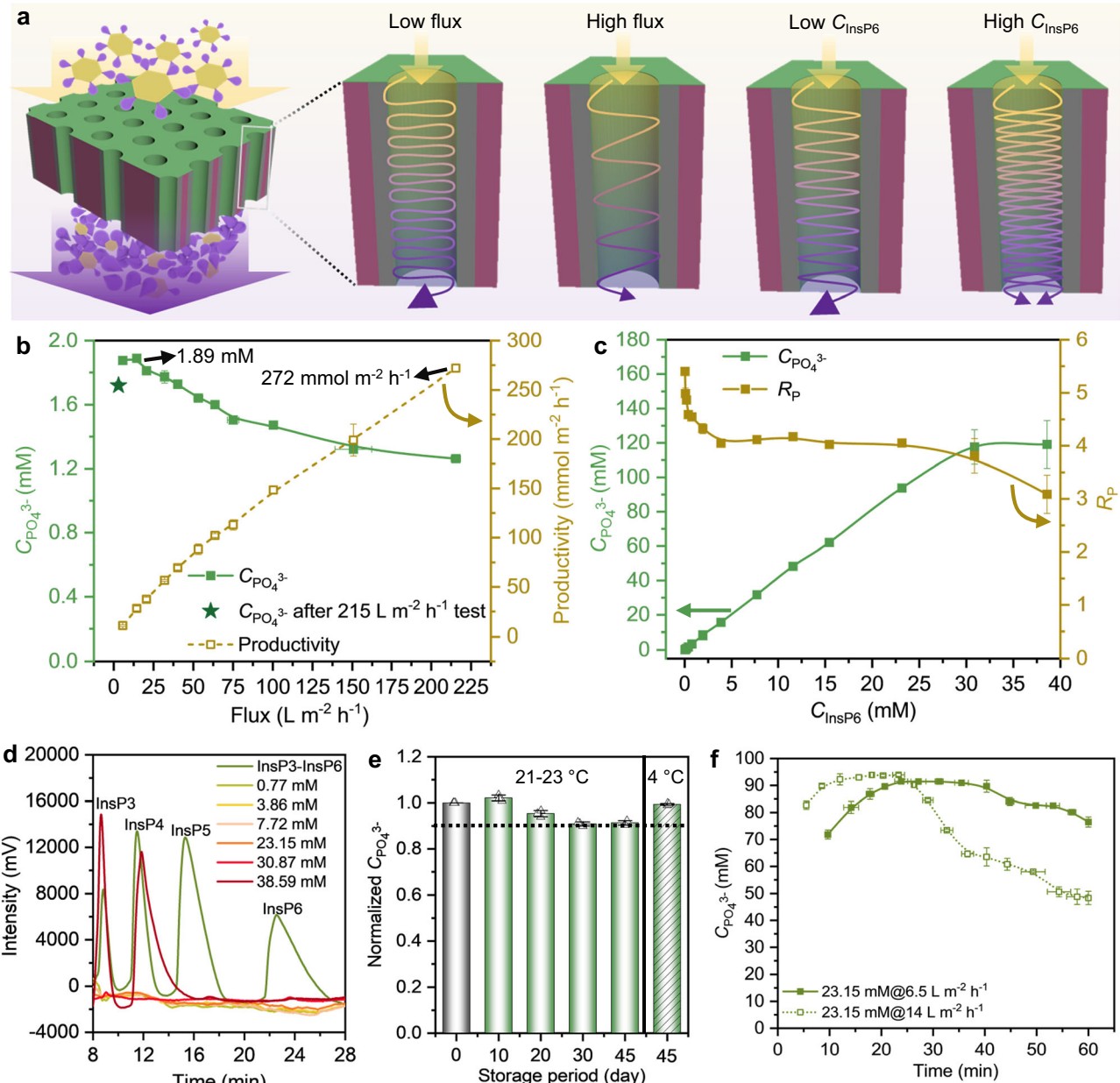

**Fig. 3 | Investigation of the catalytic performance in the continuous-flow NaMeR of YmPh-LCI@M. a** Schematic illustration of phosphate hydrolysis from phytate (myo-inositol-1,2,3,4,5,6-hexakisphophate, InsP6) by YmPh-LCI@M at a low/high flux or low/high substrate concentration. The trajectory of schematic lines illustrates the radial and longitudinal movement of liquid streams with varied concentrations of substrate InsP6, intermediate (InsP), and products. The gradient color change from yellow to violet illustrates the stepwise InsP6 hydrolysis process. The higher sinuosity of lines represents the larger number of radial collisions of the liquid stream with the nanochannels wall at a low flux. Increased line numbers represent increased substrate concentrations. The larger arrow size represents higher $R_P$ at a low flux or low substrate concentration. Parts of the diagram were adapted from ref. [20] (CC BY 4.0 DEED). **b** Effect of flux on the catalytic performance (i.e., $C_{PO_4^{3-}}$ and productivity) of YmPh-LCI@M in continuous-flow reaction with an InsP6 concentration ($C_{InsP6}$) of 0.38 mM. **c** Effect of $C_{InsP6}$ on the catalytic performance (i.e., $C_{PO_4^{3-}}$ and $R_P$) of YmPh-LCI@M under an optimal flux (~15 L m⁻² h⁻¹). **d** Composition of InsP intermediates in the hydrolysed samples under various $C_{InsP6}$, determined by reversed-phase high-performance liquid chromatography (RP-HPLC). InsP1-InsP6 standards (phytic acid solution from Sigma-Aldrich®) were used for calibration (green line). Within the standard phosphate analytics, the eluent peaks of InsP1 and InsP2 overlap, so that only InsP3-InsP6 could be separately quantified[41]. **e** Storage stability of YmPh-LCI@M: normalized $C_{PO_4^{3-}}$ with storage period in a NaOAc buffer (25 mM, pH 5.5) at room temperature (21–23 °C) or 4 °C. **f** The change of $C_{PO_4^{3-}}$ over time (operational stability) using varied fluxes in the continuous-flow mode for YmPh-LCI@M. All error bars represent s.d. of the mean from three independent experiments ($n$ = 3). Source data are provided as a Source Data file.

of the free YmPh-LCI (Supplementary Fig. 17, Supplementary Table 7). The impressive linear increase in InsP6 hydrolysis with increased $C_{InsP6}$ indicates that the surface concentration of phytases along the nanochannel within the nanoconfined environment is sufficient for InsP6 hydrolysis at comparably high substrate concentration (e.g., 23.15 mM). Additionally, YmPh-LCI@M attains its maximum $C_{PO_4^{3-}}$ of

approx. 118 mM from $C_{InsP6}$ = 30.87 mM with an approx. 64% of phosphate release efficiency (Fig. 3c, Supplementary Table 7). It indicates that a substrate saturation was reached and a further increase in substrate concentration will not increase product formation. Therefore, for the nanochannel with a length of 342 ± 23 nm with a nanochannel density of 1.44 × 10¹⁴ m⁻², an optimal

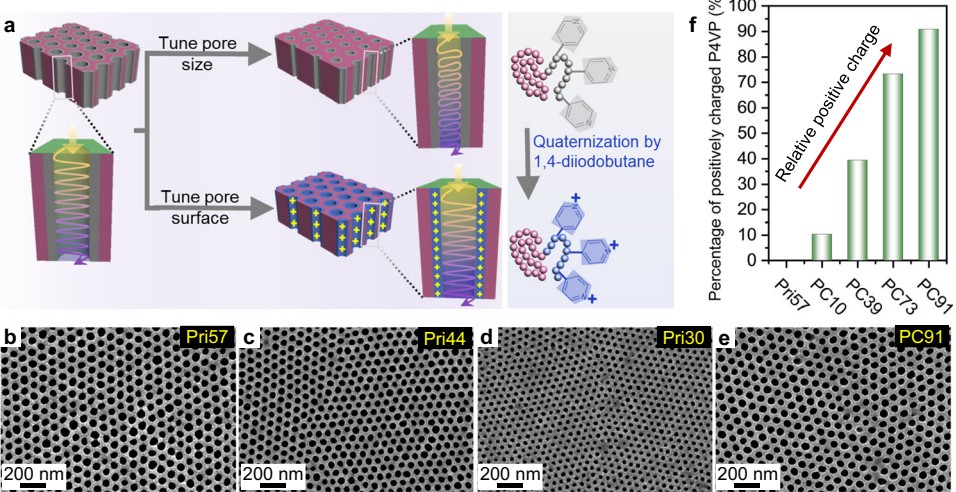

**Fig. 4 | Preparation of BCP membranes with reduced pore size or varied positively charged pore surface. a** Schematic illustration of tailoring the membrane's nanochannel properties—size and surface charge. The positive charge along the nanochannel walls is introduced via quaternization of the P4VP pore-forming block using 1,4-diiodobutane. Parts of the diagram were adapted from ref. 20 (CC BY 4.0 DEED) and ref. 29 (CC BY 3.0 DEED). **b**–**e** SEM images of the top surface of the membranes with tuned nanochannel size −57.5 nm (Pri57), 44 nm (Pri44), 30 nm (Pri30), and the representative positively charged membrane with a relative charge of 91% (PC91) derived from Pri57. Results were reproduced three times independently; representative micrographs are shown. **f** The percentage of positive charges is determined by X-ray photoelectron spectroscopy (XPS). Source data are provided as a Source Data file.

concentration of 23.15 mM should be used to maximize the phosphate yields from each InsP Intermediate.

Notably, the unique stepwise hydrolysis of InsP6 enables us to gain insight into the performance of a nanochannel as an individual nanoreactor. For this, we determined the composition of InsP intermediates via reversed-phase high-performance liquid chromatography (RP-HPLC) and then compared those results with the apparent composition calculated from $R_P$. For $R_P = 4.1$–4.6 with $C_{InsP6} = 0.77$–23.15 mM, the InsP intermediates do not reveal detectable amounts of InsP3-InsP6 compounds (Fig. 3d); products consist of InsP1 and InsP2, which matches the apparent composition with InsP1 and InsP2 calculated from $R_P$. As expected, the InsP intermediates of $R_P = 3.8$ and $C_{InsP6} = 30.87$ mM have a trace amount of InsP3, while InsP3 and InsP4 appear in the case of $R_P = 3.1$ and $C_{InsP6} = 38.59$ mM (Fig. 3c, d). Hence, the detected actual composition matches with the apparent composition calculated from $R_P$, indicating a uniform reaction from each nanochannel (individual nanoreactor). Such a uniform reaction is attributed to the uniform nanochannel diameter, homogeneous phytase immobilization, and thereby uniform nanoconfinement effects in the designed membrane reactor.

The catalytic performance is almost completely recovered after a high pressure-driven flow test of 215 L m$^{-2}$ h$^{-1}$ (star point at Fig. 3b). The latter indicates neglectable phytase detachment under high pressure-driven flow, demonstrating the excellent binding strength of YmPh-LCI@M. YmPh-LCI@M also exhibits an excellent storage stability preserving its activity up to 90% for InsP6 to InsP2/InsP1 after 45 days of storage (Fig. 3e, Supplementary Figs. 18,19). The operational stability (defined as after starting the continuous-flow reaction, the catalytic performance in terms of $C_{PO_4^{3-}}$ and $R_P$ decreases less than 10% of the maximal catalytic performance with time under certain reaction conditions) can be maintained for approx. 30 min under continuous flow (flux = 14 L m$^{-2}$ h$^{-1}$ and $C_{InsP6} = 23.15$ mM), which can be enhanced to approx. 45 min via reducing the flux to 6.5 L m$^{-2}$ h$^{-1}$ (Fig. 3f, Supplementary Fig. 20).

### Enhanced performance by tailoring BCP membrane nanochannels

To further demonstrate the potential of isoporous BCP membranes as ideal carriers, we tailored the nanochannel properties—size and surface charge (Fig. 4a) and further investigated their influence on the catalytic performance of NaMeR (e.g., $C_{PO_4^{3-}}$, $R_P$, productivity, and long-term operational stability). Specifically, the nanochannel size was reduced from a diameter of 57.5 nm to 30 nm by designing the molecular weight and composition of the PS-$b$-P4VP block copolymers, corresponding to the membranes Pri57, Pri44, Pri30 (Fig. 4b–d, Supplementary Table 1). Various percentages of positive charges from 10% to 91% were introduced along the nanochannel of the membrane Pri57 via regulating the quaternization of the P4VP pore-forming block, resulting in a series of membranes PC10, PC39, PC73, and PC91 (Fig. 4a, e, f, Supplementary Figs. 21–25)[49]. YmPh-LCI immobilized on various membranes exhibits a similar catalytic activity, highlighting that LCI is a general-applicable adhesion promoter to functionalize the nanochannel surface (Supplementary Fig. 26).

The catalytic performance is slightly increased for smaller nanochannel sizes (e.g., productivity increased from 620 mmol m$^{-2}$ h$^{-1}$ of Pri57 to 725 mmol m$^{-2}$ h$^{-1}$ of Pri30). Particularly, Pri30 possesses a desired steady catalytic performance ($C_{PO_4^{3-}} = 101$ mM, $R_P = 2.6$). This improvement can be likely attributed to an enhanced nanoconfinement effect with an increased surface concentration of the enzyme and increased contact frequency of enzymes and substrate molecules (Fig. 5a, c, Supplementary Figs. 27–29).

All positively charged membranes exhibit a steady catalytic performance (Fig. 5b, Supplementary Fig. 27), while their productivity increases with the percentage of positive charges (Fig. 5c). From PC39, the positively charged membranes show a pronouncedly increased $C_{PO_4^{3-}}$ than non-charged membranes (i.e., Pri57, Pri44, and Pri30, Fig. 5a, b, Supplementary Table 9). Supplementary Tables 9 and 10 provide the maximal concentration of phosphate obtained under steady conditions at varied InsP6 substrate concentrations for immobilized YmPh-LCI in various membranes in order to indicate efficient operational windows of the nano- and isoporous membrane reactor. In detail, we characterized the effect of $C_{InsP6}$ on the catalytic performance in the best two membrane reactors of PC73 and PC91. With increasing $C_{InsP6}$, $C_{PO_4^{3-}}$ increases strikingly (2.1 to 2.5 folds) while $R_P$ slightly decreases (Fig. 5d, e, Supplementary Figs. 30,31, Supplementary Table 10). Eventually, under the optimized conditions of $C_{InsP6} = 100$ mM and flux = 7.5 L m$^{-2}$ h$^{-1}$, PC91 possesses a catalytic performance with $C_{PO_4^{3-}} = 434$ mM, $R_P = 4.34$ and productivity =

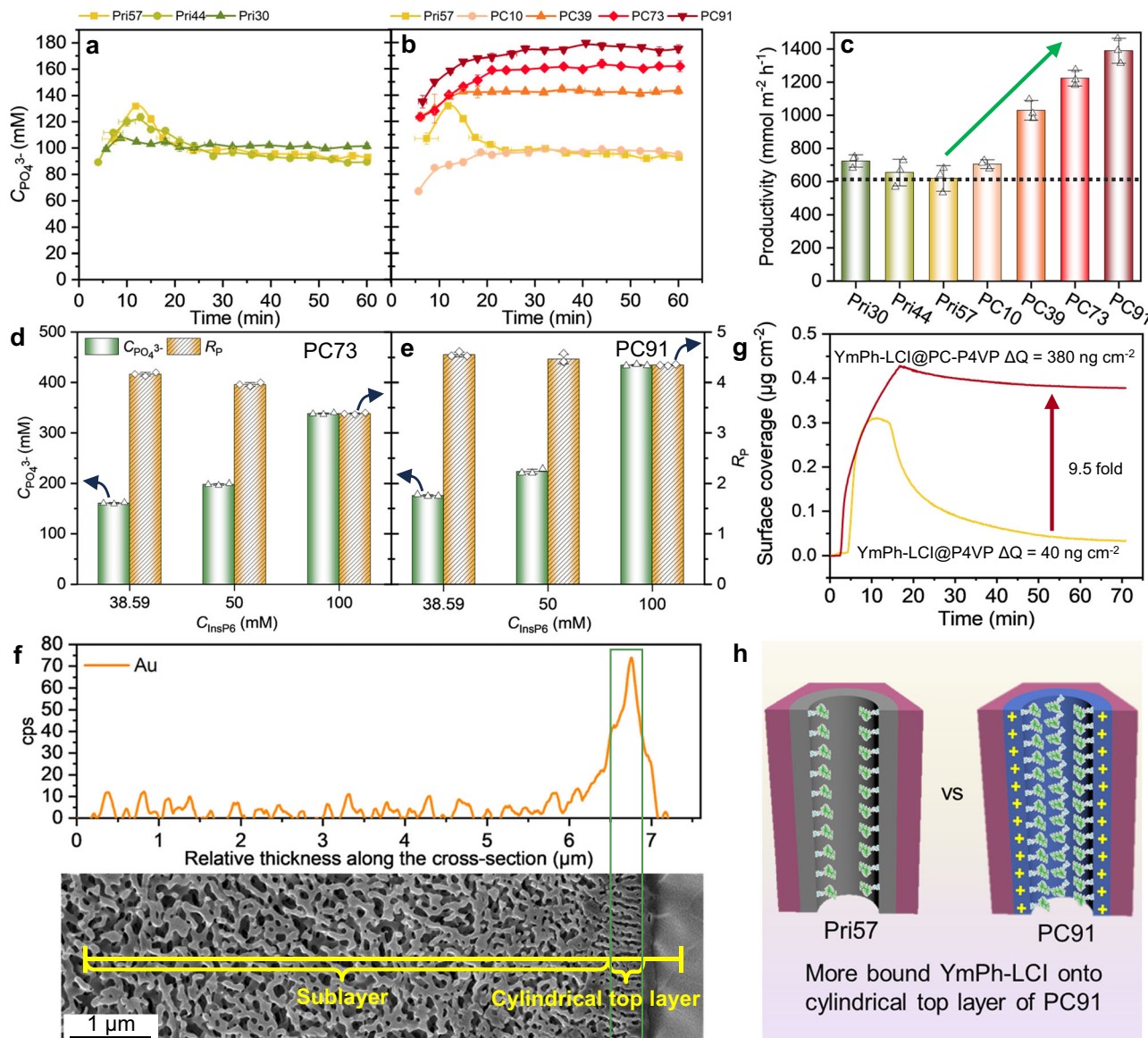

**Fig. 5 | Investigation of the catalytic performance of different continuous-flow NaMeR with tailored nanochannel properties.** The change of $C_{PO_4^{3-}}$ with time in continuous-flow reaction with $C_{InsP6}$ = 38.6 mM and flux of 6.6-7.9 L m$^{-2}$ h$^{-1}$ (Supplementary Fig. 28) for various YmPh-LCI@M immobilized membranes: **a** Pri57, Pri44, and Pri30; (**b**) Pri57, PC10, PC39, PC73, and PC91. **c** The comparison of productivity at steady state for various YmPh-LCI@M immobilized membranes Pri30, Pri44, Pri57, PC10, PC39, PC73, and PC91. **d**, **e** Effect of $C_{InsP6}$ on $C_{PO_4^{3-}}$ and $R_P$ of PC73 and of PC91 at the flux of 7.5-8.0 L m$^{-2}$ h$^{-1}$ (Supplementary Table 10). **f** A reconstructed elemental line scan of the gold signal along the cross-section of the membrane (PC91) with gold-labelled YmPh-LCI_Au from EDX mapping. The membrane's top surface faces right. Results were reproduced three times independently; representative micrograph is shown. **g** The binding affinity of YmPh-LCI towards poly(4-vinyl pyridine) (P4VP) and positively charged P4VP (PC-P4VP) thin film surfaces, determined by surface plasmon resonance (SPR) spectroscopy. **h** Schematic representation of the YmPh-LCI binding within nanochannels of the cylindrical top layer of Pri57 and PC91. All error bars represent s.d. of the mean from three independent experiments ($n$ = 3). Source data are provided as a Source Data file.

3270 mmol m$^{-2}$ h$^{-1}$ (Fig. 5e, Supplementary Table 11). Such high productivity would mean that producing 1 kg phosphate per hour would merely need approx. 3.22 m$^2$ membranes with 48.36 μmol immobilized enzyme.

Oppositely charged surface/substrate enhancing the catalytic performance of the immobilized enzyme onto a solid surface was reported, attributing to the electrostatic interaction-enhanced mass transfer in terms of substrate capture and product release[50]. In our study, given that the electrostatic attractions between the oppositely charged nanochannel wall and substrate (i.e., the positively charged nanochannel surface and negatively charged InsP6/InsP intermediates), we expect that the corresponding electrostatic attractions might prolong the contact time of substrates with the nanochannel

wall, thus enhancing hydrolysis of InsP6 and InsP Intermediates. Moreover, InsP6 with its six negatively charged phosphate groups might experience a longer averaged contact time with the positively charged nanochannel walls than InsP intermediates with fewer phosphate groups due to a stronger electrostatic attraction between InsP6 and nanochannels. A longer contact time and stepwise hydrolysis of phosphate from InsP6 might result in a steep and a stepwise local concentration gradient from the uppermost segment of the nanochannel. As a result, a high specific phytase activity was ensured for a rapid and efficient stepwise hydrolysis of InsP6 to InsP2.

Additionally, YmPh-LCI might be preferentially located at the cylindrical top layer of the nanochannel rather than in the sublayer when PC91 is compared to Pri57 (Figs. 5f, h, 2g). The latter can be

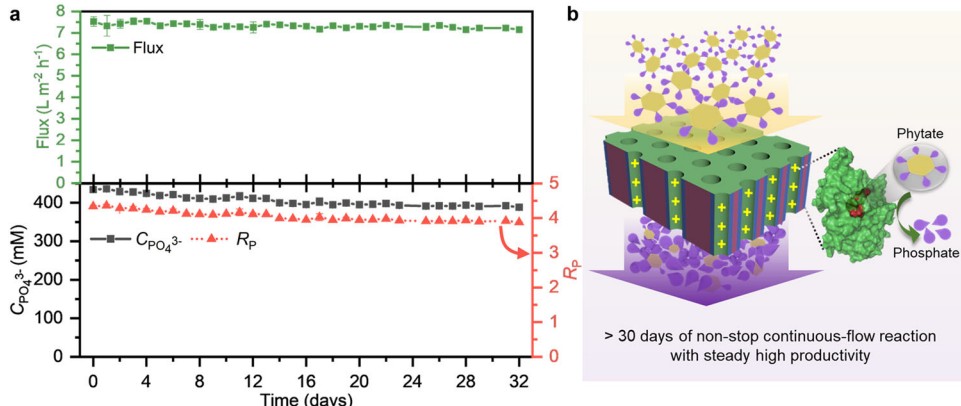

**Fig. 6 | The long-term catalytic performance of PC91 NaMeR under a high substrate concentration. a** Long-term operational stability of the continuous-flow reaction for PC91 with $C_{InsP6}$ = 100 mM and flux = 7.2–7.5 L m⁻² h⁻¹. Error bars represent s.d. of the mean from three independent experiments ($n$ = 3).

**b** Schematic representation of high-performance, non-stop continuous, and single-pass flow reaction. Parts of the diagram were adapted from ref. 20 (CC BY 4.0 DEED). Source data are provided as a Source Data file.

attributed to an increased binding affinity of YmPh-LCI on positively charged nanochannel surfaces; YmPh-LCI has a 9.5-fold increased surface coverage on the positively charged P4VP surface when compared to the pristine P4VP surface (determined by surface plasmon resonance (SPR) spectroscopy; Fig. 5g). The increased immobilization of YmPh-LCI within the cylindrical top layer might amplify the desired nanoconfinement effect and thereby boost the catalytic performance in the NaMeR. These results demonstrate the significance of designing nanochannels that match the size demands of enzymes in continuous-flow reactors.

Overall, relying on the enhanced nanoconfinement effects, both approaches of tuning nanochannel size and introducing charges onto the nanochannel surface are demonstrated by the improvement in the catalytic performance. For the studied charged substrate (InsP6/InsP intermediates), the varied positive charges result in a significantly improved phytate-hydrolysis.

To demonstrate the long-term operational stability of the designed NaMeR (i.e., PC91), we performed a non-stop continuous and single-pass phytate flow reaction for 32 days (Fig. 6, Supplementary Fig. 32). The catalytic performance preserves >95% for around 2 weeks, then gradually decreases to about 90% and levels out at 90% in the last 2 weeks.

The long-term performance of the NaMeR corresponds well to its storage stability. About 90% of activity was preserved after a storage period of one month (Fig. 3e, Supplementary Fig. 18). The similar trend of preserved activity (>90%) under continuous-flow reaction of 32 days was observed, indicating no negative effect from the additional continuous flow on the immobilized YmPh-LCI regarding conformation, activity, and binding stability. The operational stability of immobilized YmPh-LCI in PC91 was maintained up to 32 days with $C_{PO_4^{3-}}$ of about 400 mM, which is increased significantly compared to that in Pri57 maintaining up to approx. 45 min with $C_{PO_4^{3-}}$ of about 90 mM (Figs. 3f, 6). Strikingly, these results highlight that our designed NaMeR PC91 possesses excellent operational stability (>1 month) and superior catalytic performance with continuous cascade hydrolysis of at least four phosphate groups from one InsP6 in a concentrated InsP6 solution (100 mM); its productivity is up to about 2147 mol phosphate per m² membrane within 30 days and thereby maintained more than 90% of the maximal theoretical productivity of 2354 mol phosphate per m² membrane within 30 days (Supplementary Table 11). If scaled up with comparable performance it would mean that 20.4 tons of phosphate can be produced over 1 month with a 100 m² membrane. Furthermore, the membrane reactor demonstrates the space-time yield (STY) of up to 1.05 × 10⁵ g L⁻¹ d⁻¹ over 30 days with

15.02 μmol YmPh-LCI per m² membrane, which corresponds to an STY of 7.0 × 10³ g L⁻¹ d⁻¹ per μmol YmPh-LCI per m² membrane (Supplementary Table 12). Usually, STY should be >500 g L⁻¹ d⁻¹ (in case of high-amount and low-value products) or >100 g L⁻¹ d⁻¹ (in case of low-amount and high-value products), which is essential and attractive for commercial processes[51,52]. The superior catalytic performance indicates the great potential of our designed continuous-flow NaMeR for applications in green phosphate recovery.

## Discussion

A general-applicable, scalable, and robust flow reactor design concept that enables the fabrication of a high-performance, continuous-flow, and biofunctionalized NaMeR was developed and successfully validated by achieving a productivity of about 2147 mol phosphate per m² membrane within 30 days and a space-time yield of up to 1.05 × 10⁵ g L⁻¹ d⁻¹ over 30 days. Such excellent catalytic performance was achieved by using well-designed scalable nano- and isoporous BCP membranes as carriers, which bridge the size gap of nanochannels and enzymes, and an oriented phytase immobilization through an MBP (LCI). Investigation of varied nanochannel sizes and positive charge decorations within the nanochannels revealed a strong boost effect on productivity for the negatively charged substrates (i.e., InsP6 and InsP intermediates). Tuning nanochannel properties is therefore likely a key performance and general design parameter to maximize the productivity of NaMeR. In summary, we believe that the designed concept is general-applicable and scalable due to the design flexibility of the isoporous BCP membranes, and scalable/oriented immobilizations of enzymes through MBP (Supplementary Fig. 33, Supplementary Notes 4.4-4.9). The stepwise conversion of InsP6 shows that the design concept can be likely used in a multi-catalytic process when various catalysts could be immobilized within the nanochannel in a defined manner. Based on the high productivity/performance, simple operation, flexibility, and scalability in design, we envisage that the developed NaMeR concept will be of broad synthetic value for enzymatic production of bulk and fine chemicals.

## Methods

### Enzymatic activity determination by 4-MUP assay
The activity of YmPh-WT and YmPh-LCI proteins can be detected with 4-Methylumbelliferyl phosphate (4-MUP) assay[53]. The fluorescent product 4-Methylumbelliferone (4-MU) was produced by phytase hydrolysation of 4-MUP. 1 mM 4-MUP stock was prepared freshly in 250 mM NaOAc buffer (pH 5.5, 1 mM CaCl₂, 0.01% Tween-20) and kept on ice before use.

Enzyme activity determination was performed in a 100 μL reaction solution containing 4-MUP (0.5 mM) and protein (0.429 or 0.858 μM) in 250 mM NaOAc buffer (pH 5.5, 1 mM $CaCl_2$, 0.01% Tween-20) by determining the increase in relative fluorescence over time using a Tecan Infinite® M1000 microtiter plate reader (Tecan i-control version 1.35; interval: 1.5 s, time: 20 min, $\lambda_{ex}$: 322 nm, $\lambda_{em}$: 464 nm; gain: 100, room temperature). The initial reaction rate was calculated to demonstrate the enzyme activity. All experiments were performed in triplicate.

## Synthesis of diblock copolymers PS-b-P4VP

PS-b-P4VP diblock copolymers were synthesized by sequential living anionic polymerization (Supplementary Fig. 34)[49]. The solvent used for polymerization was tetrahydrofuran (THF), purified over sodium metal, and titrated with sec-butyl lithium (s-BuLi, Sigma-Aldrich, USA, 1.4 M solution in cyclohexane). Styrene was purified over a column filled with aluminum oxide and subsequently distilled from di-n-butyl magnesium (Sigma-Aldrich, USA, 1.0 M solution in heptane). 4-Vinylpyridine (4VP) was distilled under reduced pressure after being treated twice with ethylaluminium dichloride (Sigma-Aldrich, USA, 1 M in n-hexane). The polymerization of purified styrene was initiated by s-BuLi in THF at −78 °C for approximately 2 h. A 5 ml aliquot was collected from the reactor and terminated with degassed methanol for ${}^{1}H$ NMR and GPC analysis of PS. Afterwards, the purified 4VP was added to the reactor via a syringe and polymerized onto the living polystyrene for approx. 14 h. The polymerization was terminated with degassed methanol/acetic acid (90/10 by volume, v/v). After the removal of THF under reduced pressure and precipitation in water, the PS-b-P4VP diblock copolymers were obtained.

## Isoporous membrane fabrication by BCP self-assembly and non-solvent-induced phase separation (SNIPS)

Integral asymmetric isoporous membranes were prepared by using the SNIPS technique. In general, the PS-b-P4VP diblock copolymers were dissolved in a solvent mixture of THF/N, N-dimethylformamide (DMF) (40:60 weight%). After stirring for 24 h, the solutions were directly cast on a polyester nonwoven support using a doctor blade with a gap height of 200 μm. The films were left for a certain time (5-20 s) under air before immersing them in a non-solvent bath (water bath). After immersing for several hours in the water bath, the membranes were dried at 60 °C in a vacuum oven.

## Optimization of the binding condition by 4-MUP assay

The isoporous membrane was cut into small pieces with 6 mm diameter (corresponding to a membrane top surface area of $A = 0.28\ cm^2$) and then each piece was transferred into a well of a 96-well microtiter plate (MTP) (PS, F-bottom; Greiner Bio-One GmbH) ensuring the polyester layer support was attached to the MTP bottom and the top of the isoporous membrane faced up.

100 μL washing solution A (250 mM NaOAc buffer, pH 5.5, 1 mM $CaCl_2$, 0.01% Tween-20) was applied to prewash the membrane with pipetting up and down. Afterwards, the prewashed membrane was washed again with 100 μL washing solution B (250 mM NaOAc buffer, pH 5.5) twice. Subsequently, the membrane was washed with 100 μL binding solution (50 mM Tris-HCl buffer, pH 8.0) followed by 100 μL protein solution (ranging from 0 to 9 μM) diluted in the binding solution. After incubation on a MTP shaker (ELMI SkyLine DTS-4 Digital Thermo Shaker; Elminorthamerica Ltd.; 10 min, 600 rpm, 25 °C), the protein-loaded membrane was washed with 100 μL washing solution A (MTP shaker, 5 min, 600 rpm, 25 °C). The washing steps were repeated three times. Finally, the washed membrane was pinched from MTP using tweezers and dried using nitrogen gas. Membrane pieces were transferred to a new unused MTP facing up before starting the 4-MUP assay. YmPh-WT (wild-type phytase without LCI) was applied as the negative control.

The amount of immobilized enzyme (pmol $cm^{-2}$) was determined in a stepwise process (see Supplementary Table 2) by determining the starting protein concentration ($C_0$, g $L^{-1}$), supernatant concentration before washing ($C_1$, g $L^{-1}$), after first washing ($C_{w1}$, g $L^{-1}$), after second washing ($C_{w2}$, g $L^{-1}$), and after third washing ($C_{w3}$, g $L^{-1}$). Protein concentrations were determined at a wavelength of 280 nm using a UV spectrophotometer (Nanodrop™ 2000; Thermo Fisher Scientific) for YmPh-WT (47.3 kDa, 49890 $M^{-1}\ cm^{-1}$) and YmPh-LCI (55.1 kDa, 73840 $M^{-1}\ cm^{-1}$). MTP wells without membrane inside were applied as negative controls. All experiments were performed in triplicate. The amount of the immobilized enzyme on the membrane was calculated using the following equation:

$$Immobilized\ enzyme = \frac{mol\,(enzyme)}{membrane\ area}$$
$$= \frac{(C_0 - C_1 - C_{w1} - C_{w2} - C_{w3}) \times V \times 10^3}{A \times M} \quad (1)$$

where $V$ is the volume of solutions (μL), $M$ is protein molecular weight (kDa), and $A$ is the top surface area of the membrane ($cm^2$).

The binding efficiency was calculated using the following equation:

$$Binding\ efficiency = \frac{mol\,(binding\ enzyme)}{mol\,(feed\ enzyme)}$$
$$= \frac{(C_0 - C_1 - C_{w1} - C_{w2} - C_{w3}) \times V \times M}{C_0 \times V \times M} \quad (2)$$

where $V$ is the volume of solutions (μL). $M$ is the protein molecular weight (kDa).

For determination of the enzymatic activity after immobilization on the catalytic membrane, 100 μL reaction solution containing 4-MUP (0.5 mM) in washing solution A was loaded to each well of an MTP and incubated (MTP shaker, 600 rpm, 25 °C). The relative fluorescence over time (0.5, 5.5, 10.5, 15.5, 20.5, 25.5 min) was measured using a CLARIOstar® microtiter plate reader (MARS version 6.20, BMG Labtech, $\lambda_{ex}$: 322 nm, $\lambda_{em}$: 464 nm; gain: 100, room temperature). The maximum reaction rate was calculated to demonstrate the enzymatic activity of the catalytic membrane. All experiments were performed in triplicate. The reactivity (RFU $s^{-1}$ $pmol_{enzyme}^{-1}$) of free YmPh-LCI and YmPh-LCI@M was calculated using the following equation:

$$Reactivity = \frac{reaction\ rate\,(RFUs^{-1})}{mol\ of\ enzyme\,(pmol)} \quad (3)$$

Please note that the reactivity of YmPh-LCI@M does not reflect its performance since the flow through the membrane is restricted by diffusion. It can rather be regarded as a quick experiment to see whether the immobilized enzymes are active under substrate depletion and product accumulation conditions (Supplementary Table 13).

## Immobilization of YmPh-LCI onto thin dense PS-b-P4VP films

Thin dense PS-b-P4VP films were prepared via spin-coating of a 10 mg $mL^{-1}$ polymer solution in $CHCl_3$ on a cleaned silicon wafer (1 cm × 1 cm) at 3000 rpm for 2 min. The prepared films were dried at room temperature in a vacuum oven overnight to remove $CHCl_3$ residues.

Afterwards, YmPh-LCI was immobilized onto these PS-b-P4VP films. Specifically, PS-b-P4VP films were prewashed with Tris-HCl buffer (50 mM, pH 8.0), immersed into 400 μL YmPh-LCI solution (5.2 μM) in Tris-HCl buffer (50 mM, pH 8.0) at a shaking speed of 150 rpm at room temperature (20−22 °C) for 1 h. To remove the unbound or weakly bound protein molecules, the treated films were thoroughly

rinsed with Tris-HCl buffer (50 mM, pH 8.0) twice, then immersed in 2 mL Tris-HCl buffer (50 mM, pH 8.0) at a shaking speed of 150 rpm for 10 min, and again rinsed with Tris-HCl buffer (50 mM, pH 8.0) twice. To remove the effect of salt in the buffer solution on further topography measurements, the treated films were thoroughly washed with ultra-pure $H_2O$ using the washing procedure mentioned above. Afterwards, the films were dried under atmospheric pressure for further atomic force microscopy (AFM) investigations.

To probe the conformation stability of immobilized YmPh-LCI on thin dense PS-*b*-P4VP film, the prepared films with immobilized YmPh-LCI were stored in NaOAc buffer (25 mM, pH 5.5) at room temperature (20−22 °C) for 14 days. The topography before and after storage was measured by AFM.

### Fluorescent labelling of YmPh-LCI and immobilization onto isoporous membrane

YmPh-LCI was labelled with a fluorescent dye (sulfo-cyanine 3 NHS ester). Purified YmPh-LCI was buffer exchanged to PBS buffer (1×, pH 7.4) with a final concentration of 50 µM. The sulfo-cyanine 3 NHS ester powder was freshly dissolved in ddH$_2$O with a concentration of 6 mM. 1.5 mL total reaction mixture containing 1.2 mL of 50 µM enzyme, 0.06 mL of 6 mM sulfo-cyanine 3 NHS ester and 0.24 mL PBS buffer (1×, pH 7.4) with the molar ratio of enzyme to sulfo-cyanine 3 NHS ester of 1:6. The reaction was incubated in the dark for 1 h at room temperature (21.1 °C, 350 rpm), then transferred to 4 °C and incubated overnight in the dark (16−18 h). Afterwards, the solution was dialyzed against NaOAc buffer (100 mL, 25 mM, pH 5.5) for 4 days in the dark at 4 °C, and the dialysis buffer was changed every day three times. The successful conjugation of the fluorescent molecule was further confirmed by SDS-PAGE (Supplementary Fig. 8). The fluorescently labelled YmPh-LCI (YmPh-LCI_Cy3) was immobilized onto the isoporous membranes by pressure-driven filtration in a dead-end mode as shown in Supplementary Fig. 12. Fluorescently labelled enzyme solution in the Tris-HCl buffer (50 mM, pH 8.0) penetrated through the membranes using a stirred cell (Amicon® Stirred Cells 8010) with an effective membrane area of around 3.73 cm$^2$ under a transmembrane pressure of 20-30 mbar. Afterwards, the resulting membranes were thoroughly washed with NaOAc buffer (250 mM, pH 5.5) using a higher transmembrane pressure of 40-60 mbar, to remove all unbound proteins within the membrane nanostructure. This washing procedure was repeated several times until no more YmPh-LCI_Cy3 was detectable in the washing buffer solution. The whole process was done in the dark to avoid undesired photobleaching. The prepared membrane was dried under atmospheric pressure for a few hours before further measurements.

### Gold nanoparticle labelling of YmPh-LCI and immobilization onto isoporous membrane

YmPh-LCI was labelled with 1.4 nm mono-sulfo-N-hydroxysuccinimide ester-Nanogold® (Mono-Sulfo-NHS-Nanogold®, Nanoprobes, Inc. New York, USA) via the efficient reaction between primary amines and NHS group following the manufacturer's instructions. One vial of the 6 nmol Mono-Sulfo-NHS-Nanogold® was freshly dissolved in 0.2 mL deionized H$_2$O, then immediately mixed with 0.178 mL, 45 µM purified enzymes in PBS buffer (pH 7.4, 10 mM Na$_2$HPO$_4$, 1.8 mM KH$_2$PO4, 137 mM NaCl, 2.7 mM KCl). Afterwards, the mixture was incubated under shaking (150 rpm) for 2 h at room temperature (21 °C) and then transferred to 4 °C in a refrigerator overnight (16-18 h). The molar ratio of Mono-Sulfo-NHS-Nanogold® to enzyme was 3:4. Dynamic light scattering (DLS) measurement (Wyatt DYNAMICS version 4.0.1.5) was implemented to confirm the conjugation of Nanogold® to YmPh-LCI (Supplementary Fig. 9). Afterwards, Mono-Sulfo-NHS-Nanogold® labelled enzyme was immobilized onto the membrane following a similar procedure as described for the fluorescently labelled enzyme.

The enzyme solution in Tris-HCl buffer (50 mM, pH 8.0, 4 mM Triton X-100) was filtered through the membranes with a transmembrane pressure of 20-30 mbar. The permeate enzyme solution was recycled to repeat this filtration procedure 2-3 times, to provide sufficient contact between the enzyme and membrane. Afterwards, the resulting membranes were thoroughly washed with NaOAc buffer (250 mM, pH 5.5) to remove all unbound or weakly bound enzymes within the membrane nanostructure. The prepared membranes were dried at room temperature in a vacuum oven before further measurements.

### Introduction of the positive charge along the nanochannel surface

The positive charges on the nanochannel surface were introduced via quaternization of the P4VP pore-forming block using 1,4-diiodobutane (DIB)[49]. The quaternization was performed by placing the membranes into a reactor (i.e., a desiccator) with a predetermined amount of liquid DIB. The desiccator was evacuated by a diaphragm vacuum pump (Vacuubrand, Wertheim, Germany) to facilitate the formation of the DIB vapour phase. Afterwards, the evacuated desiccator was kept at room temperature for a predetermined time. The resulting membrane was taken out from the venting desiccator and transferred to a vacuum oven at 60 °C for 2 days to remove unreacted DIB.

### Characterization

The chemical composition of block copolymers was measured by [1]H NMR on a Bruker 500 MHz Avance III HD NMR spectrometer using deuterated chloroform (CDCl$_3$) as a solvent and the software TopSpin 3.2, and analysed using the software MestReNova (version 14.2.1-27684). Molecular weights and dispersity indices of the polymers were determined by gel permeation chromatography (GPC). The measurements were performed at 50 °C in THF or dimethyl-acetamide (DMAc) with LiCl using 3 µm PSS SDV gel columns at a flux of 1.0 mL min$^{-1}$ (VWR-Hitachi 2130 pump, Hitachi, Darmstadt, Germany). A Waters 2410 refractive-index detector ($\lambda = 930$ nm) with a polystyrene (PS) calibration was used. The data were collected and analysed using the software PSS® Win GPC UniChrom.

Fourier transform infrared spectroscopy (FTIR) was conducted to detect the post-functionalization reaction using a Bruker Alpha (diamond-ATR unit) with the software OPUS (version 8.2.28, Bruker, Karlsruhe, Germany). For analyzing the chemical composition of the modified membranes with 1,4-diiodobutane, X-ray photoelectron spectroscopy (XPS) was carried out by using a Kratos AXIS Ultra DLD spectrometer (Kratos, Manchester, UK) with an Al-Kα X-ray source (monochromator) operated at 225 W under vacuum of $<2.5 \times 10^{-9}$ Torr. The analyzed area was 700 µm × 300 µm. For survey spectra, a pass energy of 160 eV was employed, while for the region scans, a pass energy of 20 eV was selected. All spectra were calibrated to 284.5 eV binding energy of the C 1 s signal. For all samples, charge neutralization was necessary. The evaluation and validation of the data were accomplished with the software CASA-XPS version 2.3.18. For deconvolution of the region files, background subtraction (linear or Shirley) was performed before calculation.

The membrane morphology was investigated by scanning electron microscopy (SEM) and transmission electron microscopy (TEM). SEM images were taken on a Merlin (ZEISS, Oberkochen, Germany) at accelerating voltages of 1.5-3 kV using the software SmartSEM. Cross-sectional specimens were prepared by immersing and fracturing the membranes in liquid nitrogen, followed by sputter-coating with 1-1.5 nm platinum using a CCU-010 coating device (Safematic, Switzerland). The average pore size, porosity, and thickness of the cylindrical top layer were determined using the software Image Management System (IMS, Imagic Bildverarbeitung AG, Glattbrugg, Switzerland) on the basis of the SEM results. TEM images were taken on a

FEI Tecnai G$^2$ F20 transmission electron microscope (Thermo Fisher Scientific, USA) with an acceleration voltage of 120 kV in bright-field mode using the software TEM Imaging & Analysis (TIA) and TEM User Interface (TUI). Cross-sectional TEM samples were produced by embedding the membrane samples in an epoxy resin (EPO-TEK® 301) followed by preparing ultrathin slices (~50 nm) using a Leica Ultra-microtome EM UCT system (Leica Microsystems, Wetzlar, Germany), with a diamond knife (Diatome Ltd., Switzerland).

Energy dispersive X-ray spectroscopy (EDX) measurements were performed to determine the distribution of the Nanogold® labelled enzyme within the membrane nanostructure. For this, cross-sectional specimens were prepared by Ar-ion milling (Precisions Etching and Coating System PECS II, Gatan Inc., USA), followed by coating a 4 nm thick layer of carbon on the obtained cross-sections with the same device. EDX measurements were performed with the aforementioned SEM system, equipped with an X-Max Extreme and an X-Max 150 EDX detector (Oxford Instruments, U.K.), at a working distance of 5.6-5.7 mm, a constant magnification of 10 kx, an acceleration voltage of 5 kV, and a probe current of 100 pA.

The topography and thickness of the thin dense PS-*b*-P4VP films were measured by atomic force microscopy (AFM). The images were captured by a Bruker MultiMode 8 AFM (NanoScope IV controller, Bruker, Billerica, MA, USA) operated in PeakForce Quantitative Nano-mechanical Mapping (QNM) mode at ambient conditions, using commercial silicon SAA-HPI-SS-probes (tip radius ≈ 1 nm, spring constant ≈ 0.25 N/m, Bruker, USA) and the software NanoScope 9.2. For the film thickness measurements, the thin films were scratched with a scalpel. The height difference between the sample surface (immobilized YmPh-LCI) and silicon wafer was determined by measuring the edge of a scratch via the software Nanoscope Analysis 1.9 based on the AFM height images. Film thicknesses were determined at six different positions in the sample.

Confocal laser scanning microscopy (TSC SP8, Leica Micro-systems, Mannheim, Germany) was employed to visualize the distribution of the fluorescently labelled proteins within the membrane nanostructure. The series of high-resolution images with 1024 × 1024 pixels were captured using a 552 nm laser with 0.55% laser power, an objective of HC PL APO CS2 63x/1.4 oil, a gain of 410, and an emission wavelength of 562–580 nm. The software Leica Application Suite X (LAS X, version 3.5.2.18963) was used to quantitatively analyse the fluorescent signal along z-stack (membrane cross-section) and to reconstruct the confocal 3D images for the z-stack.

## Enzymatic reaction by pressure-driven continuous-flow

The enzymatic reaction was performed by the pressure-driven single-pass filtration in a dead-end mode. Firstly, the enzyme was immobilized by pressure-driven filtration following a similar procedure as the immobilization of the fluorescently labelled enzyme. A 2 ml enzyme solution of 5.2 µM in Tris-HCl buffer (50 mM, pH 8.0) was filtered through the membranes at a transmembrane pressure of 20-30 mbar. The permeate enzyme solution was recycled to repeat this filtration procedure 2-3 times to provide sufficient contact between the enzyme and membrane. Afterwards, the resulting membranes were thoroughly washed with NaOAc buffer (250 mM, pH 5.5) to remove all unbound or weakly bound enzymes within the membrane nanostructure.

Following the washing step, the substrate solution replaced the washing buffer solution, continuously penetrating through the membrane to perform the enzymatic reaction under a transmembrane pressure (Δ*p*) of 5-150 mbar at ambient temperature as shown in Supplementary Fig. 12. Phytic acid sodium salt from rice with a substrate purity of 94.5% ($C_6H_{18}O_{24}P_6$·xNa$^+$·yH$_2$O, Merck KGaA, Darmstadt, Germany, product code P8810) was used as a model substrate with various concentrations in NaOAc buffer (250 mM, pH 5.5). The total amount of phosphorus (P), sodium (Na), and H$_2$O in InsP6 was

determined by inductively coupled plasma atomic emission spectroscopy (High-Resolution Array ICP-OES, PlasmaQuant® PQ9000 Elite, Analytik Jena AG; software Aspect PQ version 1.2.4.0). As a result, InsP6 has a molecular formula of $C_6H_8O_{24}P_6$·5.72Na$^+$·3.53H$_2$O with a molar mass of 855.3 g mol$^{-1}$ (Supplementary Table 14). The volume change $\Delta V$ of the permeate was measured gravimetrically for time slots $\Delta t$ of 30-40 s for 30 min. The permeate samples were taken every 3-4 min for 30 min (8-10 samples were collected). The permeate flux $J_p$ was calculated as follows:

$$J_\mathrm{p} = \frac{\Delta V}{A \Delta t} \tag{4}$$

where $A$ is the top surface area of membrane.

The concentration of inorganic phosphate product ($C_{PO_4^{3-}}$) in the permeate samples was determined by a common colorimetric assay[54]. The permeate solution was mixed with an ammonium molybdate colour-developing solution (2.4 mM ammonium molybdate, 600 mM H$_2$SO$_4$, 0.6 mM antimony potassium tartrate, and 88 mM ascorbic acid) with a volume ratio of 2:1, leading to a formation of the blue complex with a UV-vis absorbance at 882 nm. The absorbance at 882 nm was measured within 2 min by a UV-vis spectrophotometer (GENESYS 10 S, Thermo Scientific). A standard curve of KH$_2$PO$_4$ with a concentration range of 0-200 µM was measured to determine $C_{PO_4^{3-}}$ (Supplementary Fig. 13).

$R_P$ is defined as the average amount of phosphate hydrolysed/released per InsP6 molecule, which was calculated using the equation:

$$R_\mathrm{P} = \frac{C_{PO_4^{3-}}}{C_{InsP6}} \tag{5}$$

where $C_{PO_4^{3-}}$ is the concentration of phosphate in the permeate solution and $C_{InsP6}$ is the concentration of substrate InsP6 in the feed solution.

Under the assumption that the maximal phosphate production is achieved if all 6 phosphate groups of InsP6 are hydrolysed, the maximal released phosphate amount is 6 times of $C_{InsP6}$. The percentage of the maximal phosphate production (termed as phosphate release efficiency in the context) is defined as the ratio between the produced phosphate amount and the maximal released phosphate amount, which was calculated using the equation:

$$\text{Phosphate release efficiency} = \frac{C_{PO_4^{3-}}}{6 \times C_{InsP6}} \times 100\% \tag{6}$$

where $C_{PO_4^{3-}}$ is the concentration of phosphate in the permeate solution and $C_{InsP6}$ is the concentration of substrate InsP6 in the feed solution.

The productivity (or molar flux of phosphate) is defined as the molar amount of phosphate produced per unit of membrane area and time, evaluated using the equation:

$$\text{Productivity} = C_{PO_4^{3-}} \times J_p \tag{7}$$

where $C_{PO_4^{3-}}$ is the concentration of phosphate in the permeate solution and $J_p$ is the permeate flux.

The compositions of the representative permeate samples were determined by the reversed-phase high-performance liquid chromatography (RP-HPLC, Nexera X2, Shimadzu Deutschland GmbH, Duisburg, Germany; Labsolution version 5.54 SP2), the detailed procedure is shown in Supplementary Methods.

## Storage stability of immobilized enzyme

To estimate the storage stability of the immobilized enzyme, the membranes with immobilized enzyme were stored in NaOAc buffer

(25 mM, pH 5.5) at 4 °C for 45 days or at room temperature for 10, 20, 30, or 45 days. Afterwards, the enzymatic reaction was performed following the above pressure-driven continuous flow using 0.5 mM InsP6 solution. The stability was determined by comparing the catalytic performance of the stored membranes and freshly prepared membranes.

**Long-term operational stability of non-stop continuous-flow reaction**

The long-term operational stability of continuous-flow enzymatic reaction was investigated using the same setup as above (pressure-driven continuous flow at ambient temperature). The InsP6 substrate solution (100 mM) continuously flows through the membrane with the immobilized enzyme at a constant flux for a predetermined time (e.g., 32 days). The permeated samples were taken at least three times per day to obtain the average value of the catalytic performance. The product concentration of phosphate was determined based on the above-mentioned colorimetric assay[54].

**Reporting summary**

Further information on research design is available in the Nature Portfolio Reporting Summary linked to this article.

## Data availability

All data supporting the findings of this study are available within the article and the Supplementary Information file, or available from the corresponding authors upon request. The source data underlying Figs. 2–6, Supplementary Figs. 1–5, 7–11, 13, 15–18, 20, 22–24, 26–28, and 30–33 are provided as a Source Data file. Source data are provided with this paper.

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

## Acknowledgements

The authors would like to thank Dr. Nico Scharnagl for XPS measurements, Sandra König and Patrick Funnemann for mercury-intrusion porosimetry measurements, Anke-Lisa Höhme for support in the TEM characterization, Dr. Heike Helmholz for support in the confocal fluorescent microscopy, Brigitte Lademann and Ivonne Ternes for polymer synthesis, Maren Brinkmann and Silvio Neumann for polymer characterization, Renate Jansen for the ICP-OES measurement, Dr. Anna Joelle Ruff and Dr. Lilin Feng for the helpful discussions, Jens Bührmann for SPR analysis, Heiko Notzke and Sebastian Tödten for the customized measurement cell, Dr. Shuaiqi Meng for protein structure preparation. Liang Gao is supported by a Ph.D. scholarship from the China Scholarship Council (CSC No.201708330280).

## Author contributions

V.A., U.S., and Z.Z conceived the idea and designed the experiments. Z.Z. performed and interpreted the BCP synthesis, fabrication and post-modifications of BCP membranes, enzyme immobilization onto membranes/films, gold nanoparticle labelling of the enzyme, continuous-flow reactions, colorimetric assay, confocal fluorescent microscopy measurements, and FTIR measurements. L.G. designed experiments, performed and interpreted the enzyme preparation, enzyme immobilization onto membranes, optimization of enzyme binding conditions, 4-MUP assay, fluorescent labelling of the enzyme, RP-HPLC measurement, DLS measurements, and SPR measurements. B.B. performed the colorimetric assay. J.S. characterized phytate substrate. L.A. established the enzyme expression protocol. E.S.S. performed the SEM and EDX measurements. E.S. performed the AFM measurements. M.H. performed the TEM measurements. Z.Z., E.S.S., E.S., and M.H. interpreted the SEM, EDX, AFM, and TEM results. T.E. performed and interpreted PFG-NMR measurements. Z.Z. and L.G. wrote the draft of the manuscript. Z.Z., L.G., V.A., and U.S. did the main discussions and manuscript editing, and A.B. and F.J. provided valuable comments. All authors contributed to the final manuscript editing.

## Funding

## Competing interests

The authors declare no competing interests.
