## [Peer Review File · Nature Communications]

An Enzymatic Continuous-Flow Reactor Based on a Pore-size Matching Nano- and Isoporous Block Copolymer MembraneEditorial Note: In this document, instances of the manuscript figure, Figure 3a, were adapted from ref 20 (CC BY 4.0 DEED). Reference 20: Zhang, Z. *et al.* Hybrid Organic–Inorganic–Organic Isoporous Membranes with Tunable Pore Sizes and Functionalities for Molecular Separation. *Adv. Mater.* 33, 2105251, doi:10.1002/adma.202105251 (2021).

REVIEWER COMMENTS

Reviewer #1 (Remarks to the Author):

The manuscript describes the development of a biocatalytic flow reactor consisting of polymeric porous membranes. The porous membrane structure of the carrier onto which the enzyme phytase is immobilized is characterized by nanoconfined channels that improve mass transfer. To increase the adsorption of the enzyme to the surface of the carrier a fusion construct with a material binding peptide (MBP) was used. The resulting biocatalyst was effective in flow characterized by a high storage and operational stability and a high space-time-yield of phosphate from phytate. This is an interesting new approach to develop immobilized enzyme catalyst specifically for continuous-flow applications. The findings are of interest in the field of biocatalysis, both for scientific and industrial applications. There are a number of comments than need to be addressed to make the manuscript suitable for publication in Nature Communications.

1. general comment: It would be nice for the reader to have the volumetric flow rates included, and not only the flux.
2. general comment: Add the space time yield per enzyme amount in the discussion or abstract. This value can be compared with other immobilized phytase biocatalysts.
3. page 12, Fig 3a: It is not clear what is exactly shown in the figure. What do the different lines represent, and on what simulation/theory is this based?
4. page 12, Fig 3b: Indicate the maximal possible phosphate concentration and the maximal productivity, which should be 1.14 mM
5. page 11-13: Indicate the percentage of the maximal phosphate production when the productivity under different conditions is reported. For example $C_{\text{phosphate}} = 118 \text{ mM}$ from $C_{\text{phytate}} = 30.87 \text{ mM}$ is circa 64% of maximal phosphate release.

6. page 15 and 16: there are several error messages, possibly broken links to references. This could be a pdf conversion error.
7. page 16, figure 5: indicate the maximal phosphate concentration that can be obtained in the different experiments in the different figures.
8. page 17: "approx. 3.22 m² membranes". Indicate the dimensions (or volume) of such a reactor and the amount of enzyme that would be immobilized on it.
9. page 18: in the positively charged material (PC91) more enzyme is immobilized as indicated by SPR (higher surface coverage). The authors argue that the nanoconfinement effect might explain the higher productivity of this catalysts, but could it just be the higher enzyme load?
10. page 19, Fig.6: how does the operational stability of PC91 compare to that of YmPh-LCI@M at the same flowrate? The authors report more than 30 day operational stability for the former and only 30 min operational stability for the latter, which only a 2-fold difference in flux. It would be useful to have volumetric flowrates as well in both cases. Indicate the maximal theoretical phosphate productivity in the figure as well. There seems to be considerable instability in the flux, can this be technically improved by using better pumps and controllers?
11. general: I find the term nanoreactor somewhat misleading as the reactor itself is not that small, and "nanoporous reactor" or "nanoporous membrane reactor" more suitable.
12. page 19-20: when comparing the STY with other processes it would also be useful to include the amount of enzyme that was used, as this would strongly contribute to the cost of the overall process.

Reviewer #2 (Remarks to the Author):

This manuscript demonstrates the fabrication of a novel isoporous continuous-flow biofunctionalized membrane nanoreactor system by integrating well-designed isoporous block copolymer membranes as carriers with genetically engineered material binding peptides fused enzyme. The uniform nanochannels of block copolymer membranes provide uniform nanoconfined environments for enzymes, together with adhesion-promoting material binding peptides, resulting in efficient and oriented phytase enzyme immobilization in isoporous nanochannels. The superior performance of the enzyme-membrane

nanoreactor was thoroughly investigated in terms of permeation flux, substrate concentration, and stability. The authors further tuned the nanochannel properties of the membranes and fortified the nanoconfinement effect, imparting enhanced catalytic performance. The resulting membrane nanoreactors have notable productivity and superior stability over 1 month in a continuous-flow process. The systematic results and important findings demonstrate the general applicability of material binding peptides, the design flexibility of block copolymer membranes, and also the significance of flexibly designing the nanochannels in the membrane nanoreactor. The whole experiments are well-designed and the manuscript is well-written, providing in-depth understanding of the enzyme binding properties, location of immobilized enzymes, and catalytic performance. The authors show for the first time the general-applicable, synergistic design approach with an enzyme-matched carrier and productive/oriented enzyme immobilization to create the high-performance catalytic continuous-flow reactor. Therefore, this work is of distinct novelty, and worth publication in Nature Communications, after well addressing the following minor issues.

1. It may be worthwhile to compare the block copolymer membranes with other nano-sized isoporous membranes explicitly, e.g, inorganic AAO membranes and track-etched polymeric membranes, making the work better understood to the broad readership of Nature Communications.

2. The authors should include the comparison of the obtained space-time-yield value with the required value of commercial processes in the abstract, to highlight their application potential.

3. In lines 155-157, the authors state that bound YmPh-LCI formed a homogeneous layer with a thickness of about 7.0 nm, which is in good agreement with the calculated size and the measured hydrodynamic diameter of the enzyme. It would be better to describe how the authors calculated the size of the enzyme.

4. In line 259, “for the nanochannel of 342 ± 23 nm with a nanochannel density of $1.44 \times 10^{14} \text{ m}^{-2}$ ”, the authors should clarify the meaning of “ 342 ± 23 nm”, which may indicate either the length or diameter of nanochannels.

5. In line 275, the authors state the excellent binding stability under high pressure-driven flow. it would be better to state it more directly and explicitly.

6. In line 295, 300, and 303, the context of “Error! Reference source not found” should be corrected.

Editorial Note: Parts of the diagram, Figure 3a, were adapted from ref 20 (CC BY 4.0 DEED). Reference 20: Zhang, Z. *et al.* Hybrid Organic–Inorganic–Organic Isoporous Membranes with Tunable Pore Sizes and Functionalities for Molecular Separation. *Adv. Mater.* 33, 2105251, doi:10.1002/adma.202105251 (2021).

Responses to Referees

We thank the reviewers for their positive, helpful, and thoughtful comments. We have responded to all the queries and comments in detail.

Reviewer #1 (Remarks to the Author):

General Comments to the Authors: The manuscript describes the development of a biocatalytic flow reactor consisting of polymeric porous membranes. The porous membrane structure of the carrier onto which the enzyme phytase is immobilized is characterized by nanoconfined channels that improve mass transfer. To increase the adsorption of the enzyme to the surface of the carrier a fusion construct with a material binding peptide (MBP) was used. The resulting biocatalyst was effective in flow characterized by a high storage and operational stability and a high space-time-yield of phosphate from phytate. This is an interesting new approach to develop immobilized enzyme catalyst specifically for continuous-flow applications. The findings are of interest in the field of biocatalysis, both for scientific and industry

al applications. There are a number of comments than need to be addressed to make the manuscript suitable for publication in Nature Communications.

1. general comment: It would be nice for the reader to have the volumetric flow rates included, and not only the flux.

Authors' response: Included. The volumetric flow rates are now in Table S6, S9, S10, S11, S12, S13 in the revised supplementary information (page 23, 28, 37-41).

We calculated the volumetric flow rate based on the flux as follows:

$$\text{Volumetric flow rate} = J_p \times A \times 10$$

where the volumetric flow rate is with a unit of mL h⁻¹; J_p is the permeate flux with a unit of L m⁻² h⁻¹; A is the top surface area of the used membrane in a unit of cm².

In our setup, we used a cell (Amicon® Stirred Cells 8010) with a constant diameter of the installed membrane of 2.5 cm. However, to fix and seal the membrane sample in the cell, one rubbery O-ring was installed above the membrane surface. During every installation, it is generally normal that the rubbery O-ring deforms to a different extent. It leads to an effective fluctuation of the membrane area in a small range from 3.53 cm² (corresponding to a diameter of 2.12 cm) to 3.94 cm² (corresponding to a diameter of 2.24 cm). So the averaged membrane area was used to calculate the volumetric flow rate from the measured flux.

In the revised supplementary information (page 23) we wrote:

“We calculated the volumetric flow rate based on the flux as the following equation:

$$\text{Volumetric flow rate} = J_p \times A \times 10$$

where the volumetric flow rate is with a unit of mL h⁻¹; J_p is the permeate flux with a unit of L m⁻² h⁻¹; A is the top surface area of the used membrane in a unit of cm².

Table S6. Effect of flux on the phytase-catalyzed hydrolysis of InsP6 for YmPh-LCI@M. $C_{\text{InsP6}} = 0.38$ mM was used for these measurements.

Pressure (mbar)	5	10	15	20	30	40	50	60	70	100	150	10 ^{a)}	-- ^{b)}
Flux (L m ⁻² h ⁻¹)	6	15	21	32	40	53	64	75	100	151	215	15	-- ^{b)}
Volumetric flow rate (mL h⁻¹)^{c)}	2.3	5.7	8.0	12.2	15.2	20.1	24.3	28.5	38.0	57.4	81.7	5.7	--^{b)}
$C_{\text{PO}_4^{3-}}$ (mM)	1.88	1.89	1.81	1.77	1.73	1.64	1.6	1.5	1.47	1.32	1.26	1.72	1.83
R_p	4.9	4.9	4.7	4.6	4.5	4.3	4.1	3.9	3.8	3.4	3.3	4.5	4.8
Phosphate release efficiency (%)	82	83	79	78	76	72	70	66	64	58	55	75	80

Productivity (mmol m ⁻² h ⁻¹)	10.64	28.15	37.41	56.73	69.57	88.32	102.14	112.95	148.45	199.06	271.95	--	--
	1)												

a) After the measurement with the flux of 215 L m⁻² h⁻¹, the InsP6 hydrolysis efficiency was determined again with a flux of 15 L m⁻² h⁻¹. b) Free YmPh-LCI in batch reaction with C_{InsP6} = 0.38 mM, 1875 pmol free YmPh-LCI at shaking with 90 rpm for 16 h, at room temperature (21-23 °C).

c) The averaged membrane area used for measurements is 3.80 cm².”

In the revised supplementary information (page 28, Table S9):

“Table S9. Effect of C_{InsP6} on the reaction efficiency of YmPh-LCI@M under optimal flux (~ 15 L m⁻² h⁻¹) and free YmPh-LCI in batch reaction (enzyme amount 1000 pmol, 18 h reaction).

C _{InsP6} (mM)	0.04	0.08	0.19	0.38	0.77	1.93	3.86	7.72	11.58	15.43	23.15	30.87	38.59
Flux (L m ⁻² h ⁻¹)	15	15	16	15	15	16	16	15	14.5	14	14.5	13	13
Volumetric													
flow rate (mL h ⁻¹) ^{a)}	5.6	5.6	5.9	5.6	5.6	5.9	5.9	5.6	5.4	5.2	5.4	4.8	4.8
C _{PO₄³⁻} (mM)	0.21	0.38	0.94	1.77	3.51	8.34	15.62	31.76	48.3	62.1	93.81	117.67	119.11
R _P	5.4	5.0	4.9	4.6	4.5	4.3	4.0	4.1	4.2	4.0	4.1	3.8	3.1
Phosphate													
release efficiency (%)	88	79	82	78	76	72	67	69	70	67	68	64	51
C _{PO₄³⁻} (mM) ^{b)}	0.22	0.40	0.95	1.80	3.53	8.66	17.71	32.74	54.13	67.69	95.25	139.17	161.54
R _P ^{b)}	5.6	5.2	4.9	4.7	4.6	4.5	4.6	4.2	4.7	4.4	4.1	4.5	4.2
Phosphate													
release efficiency (%) ^{b)}	92	83	83	79	76	75	76	71	78	73	69	75	70

a) The averaged membrane area used for measurements is 3.71 cm². b) Free YmPh-LCI batch reactions used 1000 pmol enzyme, 1 mL InsP6 solutions, and 18 h incubation at 90 rpm shaking, room temperature (21-23 °C).”

In the revised supplementary information (page 37, Table S10):

“Table S10. Comparison of maximal phosphate concentration ($C_{PO_4^{3-}}$) at steady state conditions with immobilized YmPh-LCI in membranes Pri30, Pri44, Pri57, PC10, PC39, PC73, and PC91.

Membrane	Pri30	Pri44	Pri57	PC10	PC39	PC73	PC91
$C_{PO_4^{3-}}$ (mmol L ⁻¹)	101 ± 0.7	92 ± 1.9	95 ± 1.3	97 ± 1.1	144 ± 2.7	161 ± 1.9	176 ± 2.7
Flux (L m ⁻² h ⁻¹)	7.2 ± 0.58	6.9 ± 1.12	6.6 ± 0.97	7.3 ± 0.40	7.2 ± 0.65	7.6 ± 0.82	7.9 ± 0.49
Volumetric flow rate (mL h ⁻¹) ^{a)}	2.6	2.6	2.5	2.7	2.7	2.8	2.8

All the $C_{PO_4^{3-}}$ are averaged values obtained under steady state conditions in a 1 hour continuous-flow reaction with a InsP₆ concentration (C_{InsP_6}) of 38.59 mM; see also Figure S27.a) The averaged membrane areas used for measurements is in the range of 3.59-3.80 cm²”

In the revised supplementary information (page 38 Table S11):

“Table S11. Comparison of maximal phosphate concentration ($C_{PO_4^{3-}}$) at steady state for the PC73 and PC91 with varied InsP₆ concentration (C_{InsP_6} of 38.59, 50.00, and 100.00 mM).

	PC73			PC91		
C_{InsP_6} (mmol)	38.59	50.00	100.00	38.59	50.00	100.00
$C_{PO_4^{3-}}$ (mmol L ⁻¹)	161 ± 1.9	198 ± 1.5	338 ± 3.4	176 ± 2.7	223 ± 4.2	434 ± 2.1
Flux (L m ⁻² h ⁻¹)	7.6 ± 0.82	7.9 ± 0.15	7.9 ± 0.25	7.9 ± 0.49	8.0 ± 0.40	7.5 ± 0.22
Volumetric flow rate (mL h ⁻¹) ^{a)}	2.8	2.9	2.9	2.9	2.9	2.8

All the $C_{PO_4^{3-}}$ are averaged values obtained under steady state conditions in 1 hour continuous-flow reactions. ^{a)} The averaged membrane area used for measurements is 3.68 cm² of PC73, and 3.65 cm² of PC91.”

In the revised supplementary information (page 39 Table S12):

“Table S12. The calculation of maximal theoretical productivity of PC91 within 30 days of continuous-flow reaction.

C_{InsP_6} (mmol)	$C_{PO_4^{3-}}$ (mmol L ⁻¹) ^{a)}	R_P ^{a)}	Flux J_p (L m ⁻² h ⁻¹) ^{a)}	Volumetric flow rate (mL h ⁻¹) ^{b)}	Productivity (mmol m ⁻² h ⁻¹) ^{c)}	Theoretical Productivity (mol per m ² membrane within 30 days) ^{a)}	Actual Productivity (mol per m ² membrane within 30 days) ^{e)}	Actual / Theoretical Productivity (%)
---	---------------------	---	--	--	---	--	---------------------------------------

100	434.3 ± 2.1	4.34	7.5 ± 0.22	2.7	3270	2354	2147	91.2
-----	-------------	------	------------	-----	------	------	------	------

a) $C_{PO_4^{3-}}$ and J_p are the averaged values of steady state during 1 hour continuous-flow reaction with $C_{InsP6} = 100$ mM in Figure S30. b) The averaged membrane area used for measurements is 3.61 cm². c) Productivity within 1 hour continuous-flow reaction is calculated based on the obtained $C_{PO_4^{3-}}$ and J_p at the steady state. d) The theoretical productivity within 30 days can be calculated based on the productivity within 1 hour at the steady state assuming no loss of the catalytic performance within 30 days. e) Actual productivity within 30 days is calculated based on the obtained $C_{PO_4^{3-}}$ and J_p in Figure S31.”

In the revised supplementary information (page 40 Table S13):

“Table S13. The calculation of space-time yield (STY) of immobilized YmPh-LCI of PC91 during 32 days of continuous-flow reaction.

$C_{PO_4^{3-}}$ (mmol L ⁻¹) ^{a)}	J_p (L m ⁻² h ⁻¹) ^{a)}	Volumetric flow rate (mL h ⁻¹) ^{b)}	V_r (L m ⁻²)	$M_{PO_4^{3-}}$ (g mol ⁻¹)	STY (g L ⁻¹ d ⁻¹)	Enzyme amount (μmol m ⁻²)
406 ± 14	7.3 ± 0.1	2.6	6.47 × 10 ⁻²	94.97	1.05 × 10 ⁵	15.02 ± 1.6

a) $C_{PO_4^{3-}}$ and J_p are the averaged values from 32 days of continuous-flow reaction in Fig. 6a, the main manuscript. b) The averaged membrane area used for measurements is 3.61 cm².”

2. general comment: Add the space time yield per enzyme amount in the discussion or abstract. This value can be compared with other immobilized phytase biocatalysts.

Authors' response: **Done and included.** Following the reviewer's comment, we included the amount of immobilized YmPh-LCI in PC91 in the discussion of the revised manuscript (page 21) and also in the revised supplementary information (page 40 Table S13).

In the revised manuscript (page 21):

“Furthermore, the membrane reactor demonstrates the space-time yield (STY) of up to 1.05 × 10⁵ g L⁻¹ d⁻¹ over 30 days with 15.02 μmol YmPh-LCI per m² membrane, which corresponds to an STY of 7.0 × 10³ g L⁻¹ d⁻¹ per μmol YmPh-LCI per m² membrane (Supplementary Table S13).”

In the revised supplementary information (page 40 Table S13):

“Table S13. The calculation of space-time yield (STY) of immobilized YmPh-LCI of PC91 during 32 days of continuous-flow reaction.

$C_{\text{PO}_4^{3-}}$ (mmol L ⁻¹) ^{a)}	J_p (L m ⁻² h ⁻¹) ^{a)}	Volumetric flow rate (mL h ⁻¹) ^{b)}	V_r (L m ⁻²)	$M_{\text{PO}_4^{3-}}$ (g mol ⁻¹)	STY (g L ⁻¹ d ⁻¹)	Enzyme amount ($\mu\text{mol m}^{-2}$)
406 ± 14	7.3 ± 0.1	2.6	6.47 × 10 ⁻²	94.97	1.05 × 10 ⁵	15.02 ± 1.6

a) $C_{\text{PO}_4^{3-}}$ and J_p are the averaged values from 32 days of continuous-flow reaction in Fig. 6a, the main manuscript. b) The averaged membrane area used for measurements is 3.61 cm².”

3. page 12, Fig 3a: It is not clear what is exactly shown in the figure. What do the different lines represent, and on what simulation/theory is this based?

Authors' response: **Clarified.** Fig 3a is a schematic diagram to illustrate the stepwise hydrolysis process in nanochannels under low/high flux or low/high substrate concentration. The trajectory of lines represents the radial and longitudinal movement of the liquid stream with varied concentrations of substrate, intermediates (lower phosphorylated myo-inositols, InsP), and product. The gradient color change from yellow to violet illustrates the stepwise hydrolysis process of InsP6 by phytase. The higher sinuosity of lines represents the larger number of radial collisions of the liquid stream with the nanochannels wall at a low flux. Increased line number illustrates increased substrate concentrations. The larger arrow size represents the larger amount of hydrolysed PO₄³⁻ per InsP6 (larger R_p) at a low flux or low substrate concentration.

The assumed schematic hydrolysis process of YmPh-LCI@M in Fig. 3a is based on the hydrolysis mechanism of phytase, by which phytate is stepwise hydrolyzed to lower phosphorylated myo-inositols (InsP) and released phosphates (PO₄³⁻) (*Appl. Microbiol. Biotechnol.* 103, 6435-6448; *Appl. Environ. Microbiol.* 65, 367-373). Regarding the trajectory and collision of lines in each nanochannel, it is based on the mass transfer mechanism in micro-/nano-channel reactors or membrane reactors (*Nat Nanotechnol* 2022, 17(4): 417-423; *Catal Today* 2005, 110(1-2): 15-25; *Ind. Eng. Chem. Res.* 2010, 49, 1057–1062; *Catalysts* 2020, 10(7): 725): the radial transport in transversal direction is solely from molecular diffusion while the longitudinal transport in flow direction is from both convection and molecular diffusion. The mass transport rate in terms of radial and

longitudinal transport is related to the flow rate and substrate concentration (*Nat Nanotechnol* 2022, 17(4): 417-423; *Catal Today* 2005, 110(1-2): 15-25).

To clarify this, we modified Fig. 3a and its caption in the revised manuscript (Page 11, Fig. 3a):

From the original Fig. 3a:

“

To the modified Fig. 3a:

From the original caption:

“Fig. 3. (a) Schematic illustration of phosphate hydrolysis from phytate (myo-inositol-1,2,3,4,5,6-hexakisphosphate, InsP6) by YmPh-LCI@M.”

To the modified caption:

“Fig. 1. (a) Schematic illustration of phosphate hydrolysis from phytate (myo-inositol-1,2,3,4,5,6-hexakisphosphate, InsP6) by YmPh-LCI@M at a low/high flux or low/high substrate concentration.”

The trajectory of schematic lines illustrates the radial and longitudinal movement of liquid streams with varied concentrations of substrate InsP6, intermediates (InsP), and products. The gradient color change from yellow to violet illustrates the stepwise InsP6 hydrolysis process. The higher sinuosity of lines represents the larger number of radial collisions of the liquid stream with the nanochannels wall at a low flux. Increased line numbers represent increased substrate concentrations. The larger arrow size represents higher R_P at a low flux or low substrate concentration.”

4. page 12, Fig 3b: Indicate the maximal possible phosphate concentration and the maximal productivity, which should be 1.14 mM

Authors' response: **Included**. The maximal phosphate concentration of 1.89 mM was experimentally measured by a common colorimetric assay, following the published protocol (*Anal. Biochem.*, 548, 82-90, (2018)) (The details are provided in the methods of the revised manuscript, page 31). The maximal productivity of 272 mmol m⁻² h⁻¹ was calculated by the following equation as described in the methods of the revised manuscript (page 32)

$$Productivity = C_{PO_4^{3-}} \times J_p$$

where $C_{PO_4^{3-}}$ is the concentration of phosphate in the permeate solution and J_p is the permeate flux.

Both values are included in the revised manuscript and also marked in Fig. 3b in the revised manuscript (page 11-12):

“Additionally, within a low flux range (6-15 L m⁻² h⁻¹), the YmPh-LCI@M performance shows a maximal $C_{PO_4^{3-}}$ of 1.89 mM, an R_P of 4.9, and a phosphate release efficiency of 81-82%, comparable to the maximum catalytic performance of non-immobilized/free YmPh-LCI (Fig.3b, Supplementary Fig. S14). A further increase of flux to 215 L m⁻² h⁻¹ improves the productivity sharply to the maximal value of 272 mmol m⁻² h⁻¹ due to an increased mass transfer rate (bringing more substrates into nanochannels per unit time) that may enhance specific phytase activity, while it decreases $C_{PO_4^{3-}}$ to 1.26 mM, R_P to 3.3, and the phosphate release efficiency to 55% (Fig.3b, Supplementary Table S6).”

“

”

5. page 11-13: Indicate the percentage of the maximal phosphate production when the productivity under different conditions is reported. For example $C_{\text{phosphate}} = 118 \text{ mM}$ from $C_{\text{phytate}} = 30.87 \text{ mM}$ is circa 64% of maximal phosphate release.

Authors' response: **Done**. We included the calculated values in Table S6 and Table S9 of the revised supplementary information (page 23, Table S6 and page 28, Table S9) and the corresponding discussion in the revised manuscript (page 12 and 13).

We defined the percentage of the maximal phosphate production as the term of phosphate release efficiency. Accordingly, we added the calculation method in the Methods section of revised manuscript (page 31-32), as follows:

“Under the assumption that the maximal phosphate production is achieved if all 6 phosphate groups of InsP6 are hydrolysed, the maximal released phosphate amount is 6 times of C_{InsP6} . The percentage of the maximal phosphate production (termed as phosphate release efficiency in the context) is defined as the ratio between the produced phosphate amount and the maximal released phosphate amount, which was calculated using the equation:

$$\text{Phosphate release efficiency} = \frac{C_{\text{PO}_4^{3-}}}{6 \times C_{\text{InsP6}}} \times 100\%$$

where $C_{\text{PO}_4^{3-}}$ is the concentration of phosphate in the permeate solution and C_{InsP6} is the concentration of substrate InsP6 in the feed solution.”

In the revised supplementary information (page 23, Table S6):

“Table S6. Effect of flux on the phytase-catalyzed hydrolysis of InsP6 for YmPh-LCI@M. $C_{\text{InsP6}} = 0.38$ mM was used for these measurements.

Pressure (mbar)	5	10	15	20	30	40	50	60	70	100	150	10 ^{a)}	-- ^{b)}
Flux (L m ⁻² h ⁻¹)	6	15	21	32	40	53	64	75	100	151	215	15	-- ^{b)}
Volumetric flow rate (mL h ⁻¹) ^{c)}	2.3	5.7	8.0	12.2	15.2	20.1	24.3	28.5	38.0	57.4	81.7	5.7	-- ^{b)}
$C_{\text{PO}_4^{3-}}$ (mM)	1.88	1.89	1.81	1.77	1.73	1.64	1.6	1.5	1.47	1.32	1.26	1.72	1.83
R_p	4.9	4.9	4.7	4.6	4.5	4.3	4.1	3.9	3.8	3.4	3.3	4.5	4.8
Phosphate release efficiency (%)	82	83	79	78	76	72	70	66	64	58	55	75	80

Productivity (mmol m ⁻² h ⁻¹)	10.64	28.15	37.41	56.73	69.57	88.32	102.14	112.95	148.45	199.06	271.95	--	--
	1)												

a) After the measurement with the flux of 215 L m⁻² h⁻¹, the InsP6 hydrolysis efficiency was determined with a flux of 15 L m⁻² h⁻¹ again. b) Free YmPh-LCI in batch reaction with C_{InsP6} = 0.38 mM, 1875 pmol free YmPh-LCI at shaking with 90 rpm for 16 h, at room temperature (21-23 °C).

c) The averaged membrane area used for measurements is 3.80 cm².

In the revised supplementary information (page 28, Table S9):

“Table S9. Effect of C_{InsP6} on the reaction efficiency of YmPh-LCI@M under optimal flux (~ 15 L m² h¹) and free YmPh-LCI in batch reaction (enzyme amount 1000 pmol, 18 h reaction).

C _{InsP6} (mM)	0.04	0.08	0.19	0.38	0.77	1.93	3.86	7.72	11.58	15.43	23.15	30.87	38.59
Flux (L m ⁻² h ⁻¹)	15	15	16	15	15	16	16	15	14.5	14	14.5	13	13
Volumetric flow rate (mL h⁻¹)^a	5.6	5.6	5.9	5.6	5.6	5.9	5.9	5.6	5.4	5.2	5.4	4.8	4.8
C _{PO₄³⁻} (mM)	0.21	0.38	0.94	1.77	3.51	8.34	15.62	31.76	48.3	62.1	93.81	117.67	119.11
R _P	5.4	5.0	4.9	4.6	4.5	4.3	4.0	4.1	4.2	4.0	4.1	3.8	3.1
Phosphate release efficiency (%)	88	79	82	78	76	72	67	69	70	67	68	64	51
C _{PO₄³⁻} (mM) ^b	0.22	0.40	0.95	1.80	3.53	8.66	17.71	32.74	54.13	67.69	95.25	139.17	161.54
R _P ^b	5.6	5.2	4.9	4.7	4.6	4.5	4.6	4.2	4.7	4.4	4.1	4.5	4.2
Phosphate release efficiency (%)^b	92	83	83	79	76	75	76	71	78	73	69	75	70

a) The averaged membrane area used for measurements is 3.71 cm². b) Free YmPh-LCI batch reactions used 1000 pmol enzyme, 1 mL InsP6 solutions, and 18 h incubation at 90 rpm shaking, room temperature (21-23 °C).”

In the revised manuscript, the following paragraphs were updated to refer to the percentage of maximal phosphate concentration (termed as the phosphate release efficiency), to determine the overall performance of phosphate recovery under different conditions:

In the revised manuscript (page 12):

“Additionally, within a low flux range (6-15 L m⁻² h⁻¹), the YmPh-LCI@M performance shows a maximal C_{PO₄³⁻} of 1.89 mM, a R_P of 4.9, and a phosphate release efficiency of 81-82%, comparable to the maximum catalytic performance of non-immobilized/free YmPh-LCI (Fig. 3b, Supplementary Fig. S14). A further increase of flux to 215 L m⁻² h⁻¹ improves the productivity sharply to the maximal value of 272 mmol m⁻² h⁻¹ due to an increased mass transfer rate (bringing more substrates into nanochannels per unit time) that may enhance specific phytase activity, while it decreases C_{PO₄³⁻} to 1.26 mM, R_P to 3.3, and the phosphate release efficiency to 55% (Fig. 3b, Supplementary Table S6).”

In the revised manuscript (page 13):

“With increasing C_{InsP6} up to 23.15 mM, YmPh-LCI@M displays a nearly linear increase of C_{PO₄³⁻} to 93.81 mM, a steady R_P (4.0-4.3), and a steady phosphate release efficiency (67-72%), regardless of a sharp decline of R_P in the low C_{InsP6} range (< 0.8 mM) (Fig. 3c, Supplementary Table S9). Notably, this trend is similar to that of the maximum catalytic performance of the free YmPh-LCI (Supplementary Fig. S17, Table S9). The impressive linear increase in InsP6 hydrolysis with increased C_{InsP6} indicates that the surface concentration of phytase along the nanochannel within the nanoconfined environment is sufficient for InsP6 hydrolysis at comparably high substrate concentration (e.g., 23.15 mM). Additionally, YmPh-LCI@M attains its maximum C_{PO₄³⁻} of approx. 118 mM from C_{InsP6} = 30.87 mM with an approx. 64% of phosphate release efficiency (Fig 3c, Supplementary Table S9)”

6. page 15 and 16: there are several error messages, possibly broken links to references. This could be a pdf conversion error.

Authors' response: **Done**, thank you. The correct links of Figures were added in the revised manuscript (page 16):

“To further demonstrate the potential of isoporous BCP membranes as ideal carriers, we tailored the nanochannel properties – size and surface charge (Fig. 4a) and further investigated their influence on the catalytic performance of NaMeR (e.g., C_{PO₄³⁻}, R_P, productivity, and long-term operational stability). Specifically, the nanochannel size was reduced from a diameter of 57 nm to 30 nm by

designing the molecular weight and composition of the PS-*b*-P4VP block copolymers, corresponding to the membranes Pri57, Pri44, Pri30 (Fig. 4c-e, Supplementary Table S2, Fig. S25). Various percentages of positive charges from 10% to 91% were introduced along the nanochannel of the membrane Pri57 via regulating the quaternization of the P4VP pore-forming block, resulting in a series of membranes PC10, PC39, PC73, and PC91 (Fig. 4a, b, f, Supplementary Fig. S21-25)⁴⁹. ”

7. page 16, figure 5: indicate the maximal phosphate concentration that can be obtained in the different experiments in the different figures.

Authors' response: **Indicated and included** in Table S10 and Table S11 of the revised supplementary information; we agree that it helps to access the operational window of the nano- and isoporous membrane reactor.

In the revised supplementary information (page 37, Table S10):

“Table S10. Comparison of maximal phosphate concentration ($C_{PO_4^{3-}}$) at steady state conditions with immobilized YmPh-LCI in membranes Pri30, Pri44, Pri57, PC10, PC39, PC73, and PC91.

Membrane	Pri30	Pri44	Pri57	PC10	PC39	PC73	PC91
$C_{PO_4^{3-}}$ (mmol L ⁻¹)	101 ± 0.7	92 ± 1.9	95 ± 1.3	97 ± 1.1	144 ± 2.7	161 ± 1.9	176 ± 2.7
Flux (L m ⁻² h ⁻¹)	7.2 ± 0.58	6.9 ± 1.12	6.6 ± 0.97	7.3 ± 0.40	7.2 ± 0.65	7.6 ± 0.82	7.9 ± 0.49
Volumetric flow rate (mL h ⁻¹) ^{a)}	2.6	2.6	2.5	2.7	2.7	2.8	2.8

All the $C_{PO_4^{3-}}$ are averaged values obtained under steady state conditions in a 1 hour continuous-flow reaction with a InsP6 concentration (C_{InsP6}) of 38.59 mM; see also Figure S27.a) The averaged membrane areas used for measurements is in the range of 3.59-3.80 cm²”

In the revised supplementary information (page 38, Table S11):

“Table S11. Comparison of maximal phosphate concentration ($C_{PO_4^{3-}}$) at steady state for the PC73 and PC91 with varied InsP6 concentration (C_{InsP6} of 38.59, 50.00, and 100.00 mM).

	PC73	PC91

C_{InsP6} (mmol)	38.59	50.00	100.00	38.59	50.00	100.00
$C_{\text{PO}_4^{3-}}$ (mmol L ⁻¹)	161 ± 1.9	198 ± 1.5	338 ± 3.4	176 ± 2.7	223 ± 4.2	434 ± 2.1
Flux (L m ⁻² h ⁻¹)	7.6 ± 0.82	7.9 ± 0.15	7.9 ± 0.25	7.9 ± 0.49	8.0 ± 0.40	7.5 ± 0.22
Volumetric flow rate (mL h ⁻¹) ^{a)}	2.8	2.9	2.9	2.9	2.9	2.8

All the $C_{\text{PO}_4^{3-}}$ are averaged values obtained under steady state conditions in 1 hour continuous-flow reactions. ^{a)} The averaged membrane area used for measurements is 3.68 cm² of PC73, and 3.65 cm² of PC91. ”

In the revised manuscript, the following paragraph was updated to refer to the Tables in order to indicate efficient operational windows of the nano- and isoporous membrane reactor (page 18):

“Remarkably, all positively charged membranes exhibit a steady catalytic performance (Fig.5b, Supplementary Fig. S27), while their productivity increases with the percentage of positive charges (Fig.5c). From PC39, the positively charged membranes show a pronouncedly increased $C_{\text{PO}_4^{3-}}$ than non-charged membranes (i.e., Pri57, Pri44, and Pri30, Supplementary Table S10 and Table S11). Tables S10 and S11 provide the maximal concentration of phosphate obtained under steady conditions at varied InsP6 substrate concentrations for immobilized YmPh-LCI in various membranes in order to indicate efficient operational windows of the nano- and isoporous membrane reactor. In detail, we characterized the effect of C_{InsP6} on the catalytic performance in the best two membrane reactors of PC73 and PC91. With increasing C_{InsP6} , $C_{\text{PO}_4^{3-}}$ increases strikingly (2.1 to 2.5 folds) while R_p slightly decreases (Fig.5d, e, Supplementary Fig. S29-30, and Table S11).”

8. page 17: "approx. 3.22 m² membranes". Indicate the dimensions (or volume) of such a reactor and the amount of enzyme that would be immobilized on it.

Authors' response: **Dimensions/Value indicated.** The membrane reactor is the thin porous flat sheet membrane with a total thickness of 170-180 μm, like a non-woven fabric. If we only consider one piece of flat sheet membrane in round shape, for approx. 3.22 m²

membranes, the diameter of membrane reactor would be approx. 2.025 m while the thickness is 170-180 μm . However, the membrane has an advantage — it can be cut into various shape, stacked and integrated into flat sheet membrane modules to dramatically save the occupied space in the industrial application, which is widely used in the membrane separation technology. To obtain a membrane reactor with approx. 3.22 m^2 membranes, its exact dimension depends on the design of modules, e.g., the size of support frame, the size of spacer, and packing density and the design of modules may have a significant influence on the catalytic performance of the reactor. Having this said, in a simple calculation using a simple cross-flow membrane setup with staged sheets such as TTF cassette from Sartorius with a surface area of 0.6 m^2 (<https://www.sartorius.com/en/products/process-filtration/tangential-flow-filtration/tff-cassettes>) would suggest that stacking of five to six TTF cassettes (approx. size 75 cm x 80 cm) of such small devices in a standard holder would be sufficient in an ideal case.

Based on the measurement of the amount of immobilized YmPh-LCI in PC91 (15.02 \pm 1.6 $\mu\text{mol m}^{-2}$, Supplementary Table S2), the enzyme amount in the reactor with approx. 3.22 m^2 membranes is calculated as 48.36 μmol .

Following the reviewer's comment, we included the amount of immobilized enzyme for the reactor with approx. 3.22 m^2 membranes in the revised manuscript (page 18), however we would prefer to not include a specific size demands due to the various design options of enzymatic membrane reactors:

“Such high productivity would mean that producing 1 kg phosphate per hour would merely need approx. 3.22 m^2 membranes with 48.36 μmol immobilized enzyme.”

9. page 18: in the positively charged material (PC91) more enzyme is immobilized as indicated by SPR (higher surface coverage). The authors argue that the nanoconfinement effect might explain the higher productivity of this catalysts, but could it just be the higher enzyme load?

Authors' response: **No**. This point is addressed in the revised manuscript. The amount of immobilized enzyme in PC91 membrane is about 1502 pmol cm^{-2} , only 1.8 times more

than that in Pri57 (830 pmol cm⁻²). 1.8-fold increase in the amount of immobilized enzyme is much lower than 9.5-fold obtained by SPR, since SPR measured the spin-coated flat dense films made of homopolymer poly(4-vinyl pyridine) onto gold-chips, instead of the nanoporous membrane made of block copolymer polystyrene-block-poly(4-vinyl pyridine). The phosphate concentration $C_{PO_4^{3-}}$ of Pri57 (YmPh-LCI@M) reached to the plateau with increasing the substrate concentration C_{InsP6} . The maximal $C_{PO_4^{3-}}$ is approx. 118 mM when C_{InsP6} is over 30.87 mM (Fig. 3c in the revised manuscript and Figure S17, Table S9 in the revised supplementary information). In contrast, $C_{PO_4^{3-}}$ of PC91 increased sharply with increasing C_{InsP6} from 38.59 to 100 mM as shown in Fig. 5e of the revised manuscript. It is obvious that PC91 did not reach the substrate saturation yet. Even so, PC91 exhibits a $C_{PO_4^{3-}}$ of 434 mM for $C_{InsP6} = 100$ mM, that is approx. 3.7 times more than that of Pri57. Thus, it proves that 1.8 times more immobilized enzyme is not the only contributor to the improvement of catalytic performance in PC91 membrane reactor. From our investigation, three factors together contribute to the excellent catalytic performance of PC91, i.e., prolonged contact time, more immobilized enzyme in the cylindrical top layer, and nanoconfinement effect, which has been included in the revised manuscript (page 19).

It is stated:

“Oppositely charged surface/substrate enhancing the catalytic performance of the immobilized enzyme onto a solid surface was reported⁵⁰. Given that the electrostatic interactions between the oppositely charged nanochannel wall and substrate (i.e., the positively charged nanochannel surface and negatively charged InsP6/InsP intermediates), we assumed that the contact time of substrates with nanochannel wall might be prolonged, thus enhancing hydrolysis of InsP6 and InsP-Intermediates by immobilized phytase. Moreover, InsP6 with its six negatively charged phosphate groups might experience a longer averaged contact time with the positively charged nanochannel walls than InsP intermediates with fewer phosphate groups. A longer contact time and stepwise hydrolysis of phosphate from InsP6 hydrolysis might result in a steep and a stepwise local concentration gradient from the uppermost segment of the nanochannel. As a result, a high specific phytase activity was ensured for a rapid and efficient stepwise hydrolysis of InsP6 to InsP2.

Additionally, YmPh-LCI might be preferentially located at the cylindrical top layer of the nanochannel rather than in the sublayer when PC91 is compared to Pri57 (Fig. 5f, h and Fig. 2g).

The latter can be attributed to an increased binding affinity of YmPh-LCI on positively charged surfaces; YmPh-LCI has a 9.5-fold increased surface coverage on the positively charged P4VP surface when compared to the pristine P4VP surface (determined by surface plasmon resonance (SPR) spectroscopy; Fig. 5g). The increased immobilization of YmPh-LCI within the cylindrical top layer might amplify the desired nanoconfinement effect and thereby boosts the catalytic performance in the NaMeR. These results demonstrate the significance of designing nanochannels that match the size demands of enzymes in continuous-flow reactors.”

10. page 19, Fig.6: how does the operational stability of PC91 compare to that of YmPh-LCI@M at the same flowrate? The authors report more than 30 day operational stability for the former and only 30 min operational stability for the latter, which only a 2-fold difference in flux. It would be useful to have volumetric flowrates as well in both cases. Indicate the maximal theoretical phosphate productivity in the figure as well. There seems to be considerable instability in the flux, can this be technically improved by using better pumps and controllers?

Authors' response: **Volumetric flow rates and maximal theoretical phosphate productivity are included.**

Detailed answers are as follows:

Concerning the comment: “page 19, Fig.6: how does the operational stability of PC91 compare to that of YmPh-LCI@M at the same flowrate? The authors report more than 30 day operational stability for the former and only 30 min operational stability for the latter, which only a 2-fold difference in flux.”

The comparisons of the operational stability of PC91 and YmPh-LCI@M in Pri57 membrane at the same flow rate were included in Fig. 5b, Fig. 3f, and Fig. 6a of the revised manuscript. In detail, in Fig. 5b, PC91 shows a desired steady catalytic performance while Pri57 shows a decreased catalytic performance after approx. 10-15 min in 1 hour continuous-flow reaction with a comparable flow rate. In Fig. 3f and Fig. 6a, with a comparable flux, YmPh-LCI@M shows approx. 45 min operational stability at a flux of 6.5

L m⁻² h⁻¹ (Fig. 3f) whereas PC91 exhibits a more than 30 days' operational stability at a flux of 7.2-7.5 L m⁻² h⁻¹. For YmPh-LCI@M, the operational stability is merely changed from approx. 45 min to approx. 30 min with increasing the flux by 2 times to 14 L m⁻² h⁻¹ (Fig. 3f). Therefore, the significant differences in the operational stability between PC91 and YmPh-LCI@M at a comparable flux or at fluxes with 2-fold difference are not the results of the difference in flux.

We included the operational stability of YmPh-LCI@M at a flux comparable to that of PC91 and compared them:

in the revised manuscript (page 15).

“The operational stability (defined as after starting the continuous-flow reaction, the catalytic performance in the terms of $C_{\text{PO}_4^{3-}}$ and R_{P} decreases less than 10% of the maximal catalytic performance with time under the certain reaction conditions) can be maintained for approx. 30 min under continuous flow (flux = 14 L m⁻² h⁻¹ and $C_{\text{InSP6}} = 23.15$ mM), which can be enhanced to approx. 45 min via reducing the flux to 6.5 L m⁻² h⁻¹ (Fig. 3f, Supplementary Fig. S20)”

in the revised manuscript (page 20)

“The long-term performance of the NaMeR corresponds well to its storage stability. About 90% of activity was preserved after a storage period of one month (Fig. 3e, Supplementary Fig. S18). The similar trend of preserved activity (>90%) under continuous-flow reaction of 32 days was observed, indicating no negative effect from the additional continuous flow on the immobilized YmPh-LCI regarding conformation, activity, and binding stability. The operational stability of immobilized YmPh-LCI in PC91 was maintained up to 32 days with $C_{\text{PO}_4^{3-}}$ of about 400 mM, which is increased significantly compared to that in Pri57 maintaining up to approx. 45 min with $C_{\text{PO}_4^{3-}}$ of about 90 mM (Fig. 3f, Fig. 6).”

Concerning the comment: “It would be useful to have volumetric flowrates as well in both cases.”

The volumetric flow rates are **included** in the caption of Figure S20, Table S10, and Table S13 in the revised supplementary information (page 31 and 37).

The calculation method is following the equation shown in the answer to comment 1.

In the revised supplementary information (page 31, Figure S20)

“Figure S20. The change of R_p with time using different fluxes in continuous-flow reaction for YmPh-LCI@M. The corresponding volumetric flow rates are 2.4 and 5.2 ml h⁻¹, respectively. The averaged membrane area is approx. 3.72 cm². Error bars represent s.d. of the mean.”

In the revised supplementary information (page 37, Table S10)

“Table S10. Comparison of maximal phosphate concentration ($C_{PO_4^{3-}}$) at steady state conditions with immobilized YmPh-LCI in membranes Pri30, Pri44, Pri57, PC10, PC39, PC73, and PC91.

Membrane	Pri30	Pri44	Pri57	PC10	PC39	PC73	PC91
$C_{PO_4^{3-}}$ (mmol L ⁻¹)	101 ± 0.7	92 ± 1.9	95 ± 1.3	97 ± 1.1	144 ± 2.7	161 ± 1.9	176 ± 2.7
Flux (L m ⁻² h ⁻¹)	7.2 ± 0.58	6.9 ± 1.12	6.6 ± 0.97	7.3 ± 0.40	7.2 ± 0.65	7.6 ± 0.82	7.9 ± 0.49
Volumetric flow rate (mL h ⁻¹) ^{a)}	2.6	2.6	2.5	2.7	2.7	2.8	2.8

All the $C_{PO_4^{3-}}$ are averaged values obtained under steady state conditions in a 1 hour continuous-flow reaction with a InsP6 concentration (C_{InsP6}) of 38.59 mM; see also Figure S27.a) The averaged membrane areas used for measurements is in the range of 3.59-3.80 cm².”

In the revised supplementary information (page 40, Table S13)

“Table S13. The calculation of space-time yield (STY) of immobilized YmPh-LCI of PC91 during 32 days of continuous-flow reaction.

$C_{PO_4^{3-}}$ (mmol L ⁻¹) ^{a)}	J_p (L m ⁻² h ⁻¹) ^{a)}	Volumetric flow rate (mL h ⁻¹) ^{b)}	V_r (L m ⁻²)	$M_{PO_4^{3-}}$ (g mol ⁻¹)	STY (g L ⁻¹ d ⁻¹)	Enzyme amount (μmol m ⁻²)
406 ± 14	7.3 ± 0.1	2.6	6.47 × 10 ⁻²	94.97	1.05 × 10 ⁵	15.02 ± 1.6

^{a)} $C_{PO_4^{3-}}$ and J_p are the averaged values from 32 days of continuous-flow reaction in Fig. 6a, the main manuscript. ^{b)} The averaged membrane area used for measurements is 3.61 cm².”

Concerning the comment: “Indicate the maximal theoretical phosphate productivity in the figure as well.”

We included the calculation of the maximal theoretical productivity within 30 days and also made the comparison with the actual productivity within 30 days experimentally obtained by long-term non-stop continuous flow reaction of 32 days in the revised manuscript (page 21) and revised supplementary information (page 39 Table S12).

We demonstrated that within 1 hour continuous flow reaction, PC91 exhibits a steady catalytic performance with $C_{\text{PO}_4^{3-}} = 434 \text{ mM}$, $R_p = 4.34$, and productivity = $3270 \text{ mmol m}^{-2} \text{ h}^{-1}$ under the condition of $C_{\text{InsP6}} = 100 \text{ mM}$ and flux = $7.53 \text{ L m}^{-2} \text{ h}^{-1}$ in Fig. 5e of the revised manuscript and Figure S30 of the revised supplementary information. Under the assumption of no loss of catalytic performance over 30 days of continuous flow reaction, we can calculate the maximal theoretical productivity within 30 days by the following equation:

$$\textit{Theoretical Productivity within 30 days} = C_{\text{PO}_4^{3-}} \times J_p \times 24 \times 30$$

where $C_{\text{PO}_4^{3-}}$ is the concentration of phosphate in the permeate solution with a unit of mM and J_p is the permeate flux with a unit of $\text{L m}^{-2} \text{ h}^{-1}$ at the steady state within 1 hour continuous flow reaction.

In the revised manuscript (page 21), it is stated:

“Strikingly, these results highlight that our designed NaMeR PC91 possesses excellent operational stability (> 1 month) and unprecedented catalytic performance with continuous cascade hydrolysis of at least four phosphate groups from one InsP6 in a concentrated InsP6 solution (100 mM); its productivity is up to 2147 mol phosphate per m^2 membrane within 30 days and thereby maintained more than 90 % of the maximal theoretical productivity of 2354 mol phosphate per m^2 membrane within 30 days (Supplementary Table S12).”

In the revised supplementary information (page 39, Table S12), it is stated:

“Under the assumption of no loss of catalytic performance over 30 days continuous flow reaction, the maximal theoretical productivity within 30 days was calculated by the following equation:

$$\text{Theoretical Productivity within 30 days} = C_{\text{PO}_4^{3-}} \times J_p \times 24 \times 30$$

where $C_{\text{PO}_4^{3-}}$ is the concentration of phosphate in the permeate solution with a unit of mM and J_p is the permeate flux with a unit of $\text{L m}^{-2} \text{h}^{-1}$ at the steady state within 1 hour continuous flow reaction.

Table S12. The calculation of maximal theoretical productivity of PC91 within 30 days of continuous-flow reaction.

C_{InsP6} (mmol)	$C_{\text{PO}_4^{3-}}$ (mmol L^{-1}) ^{a)}	R_p ^{a)}	Flux J_p (L m^{-2} h^{-1}) ^{a)}	Volumetric flow rate (mL h^{-1}) ^{b)}	Productivity ($\text{mmol m}^{-2} \text{h}^{-1}$) ^{c)}	Theoretical Productivity (mol per m^2 membrane within 30 days) ^{d)}	Actual Productivity (mol per m^2 membrane within 30 days) ^{e)}	Actual / Theoretical Productivity (%)
100	434.3 ± 2.1	4.34	7.5 ± 0.22	2.7	3270	2354	2147	91.2

^{a)} $C_{\text{PO}_4^{3-}}$ and J_p are the averaged values of steady state during 1 hour continuous-flow reaction with $C_{\text{InsP6}} = 100$ mM in Figure S30. ^{b)} The averaged membrane area used for measurements is 3.61 cm^2 . ^{c)} Productivity within 1 hour continuous-flow reaction is calculated based on the obtained $C_{\text{PO}_4^{3-}}$ and J_p at the steady state. ^{d)} The theoretical productivity within 30 days can be calculated based on the productivity within 1 hour at the steady state assuming no loss of the catalytic performance within 30 days. ^{e)} Actual productivity within 30 days is calculated based on the obtained $C_{\text{PO}_4^{3-}}$ and J_p in Figure S31.”

Concerning the comment: “There seems to be considerable instability in the flux, can this be technically improved by using better pumps and controllers?”

Technically we cannot see the improvement by changing the controllers. In our setup, the pressure supply was from a high pressurized gas line instead of peristaltic or membrane pumps as illustrated in Figure S12 of the revised supplementary information. The applied pressure value in Fig. 6 was in a range of 12-15 mbar, which is close to the lower limit of

control of pressure regulators. We have applied precise pressure regulators (values) and pressure gauge to have the optimal control over pressure value. Therefore, we do not see that changing pressure regulator and gauges would be highly beneficial, however including an online measurement of phosphate production to regulate flow rates might be useful for future and scale-up developments.

11. general: I find the term nanoreactor somewhat misleading as the reactor itself is not that small, and "nanoporous reactor" or "nanoporous membrane reactor" more suitable.

Authors' response: We thank the reviewer for this thoughtful comment that we like. We **included** the suggested term "nanoporous membrane reactor" in the introduction at the goal of the manuscript and the term "NaMeR" as short cut of it within the following text.

In the revised manuscript (**page 5**), it is stated:

"...**The enzyme-functionalized nano- and isoporous block copolymer membrane reactor operates at continuous flow with high-performance which we termed as "nanoporous membrane reactor" (NaMeR) in the following text.**"

Also we changed the title from

"Biofunctionalized Isoporous Continuous-flow Membrane Nanoreactor"

to

"**An Enzymatic Continuous-Flow Reactor Based on a Pore-size Matching Nano- and Isoporous Block Copolymer Membrane**".

"Biofunctionalized" was replaced by "Enzymatic" to specify the biofunction of the enzyme. Adding "Pore-size Matching Nano- and Isoporous" is to specifically emphasize the significance of designing the nanochannel size for excellent catalytic performance in continuous-flow reactors.

12. page 19–20: when comparing the STY with other processes it would also be useful to include the amount of enzyme that was used, as this would strongly contribute to the cost of the overall process.

Authors' response: **DONE**. Following the reviewer's comment, we included the enzyme amount of immobilized YmPh-LCI in PC91 in the discussion of the revised manuscript (page 21) and also in the revised supplementary information (page 40, Table S13).

In the revised manuscript (page 21):

“Furthermore, the membrane reactor demonstrates the space-time yield (STY) of up to 1.05×10^5 g L⁻¹ d⁻¹ over 30 days with 15.02 μmol YmPh-LCI per m² membrane, which corresponds to an STY of 7.0×10^3 g L⁻¹ d⁻¹ per μmol YmPh-LCI per m² membrane (Supplementary Table S13).”

In the revised supplementary information (page 40, Table S13):

“Table S13. The calculation of space-time yield (STY) of immobilized YmPh-LCI of PC91 during 32 days of continuous-flow reaction.

$C_{\text{PO}_4^{3-}}$ (mmol L ⁻¹) ^{a)}	J_p (L m ⁻² h ⁻¹) ^{a)}	Volumetric flow rate (mL h ⁻¹) ^{b)}	V_r (L m ⁻²)	$M_{\text{PO}_4^{3-}}$ (g mol ⁻¹)	STY (g L ⁻¹ d ⁻¹)	Enzyme amount (μmol m ⁻²)
406 ± 14	7.3 ± 0.1	2.6	6.47 × 10 ⁻²	94.97	1.05 × 10 ⁵	15.02 ± 1.6

^{a)} $C_{\text{PO}_4^{3-}}$ and J_p are the averaged values from 32 days of continuous-flow reaction in Fig. 6a of the main manuscript. ^{b)} The averaged membrane area used for measurements is 3.61 cm².”

Reviewer #2 (Remarks to the Author):

General Comments to the Authors: This manuscript demonstrates the fabrication of a novel isoporous continuous-flow biofunctionalized membrane nanoreactor system by integrating well-designed isoporous block copolymer membranes as carriers with genetically engineered material binding peptides fused enzyme. The uniform nanochannels of block copolymer membranes provide uniform nanoconfined environments for enzymes, together with adhesion-promoting material binding peptides, resulting in efficient and oriented phytase enzyme immobilization in isoporous nanochannels. The superior performance of the enzyme-membrane nanoreactor was thoroughly investigated in terms of permeation flux, substrate concentration, and stability. The authors further tuned the nanochannel properties of the membranes and fortified the nanoconfinement effect, imparting enhanced catalytic performance. The resulting membrane nanoreactors have notable productivity and superior stability over 1 month in a continuous-flow process. The systematic results and important findings demonstrate the general applicability of material binding peptides, the design flexibility of block copolymer membranes, and also the significance of flexibly designing the nanochannels in the membrane nanoreactor. The whole experiments are well-designed and the manuscript is well-written, providing in-depth understanding of the enzyme binding properties, location of immobilized enzymes, and catalytic performance. The authors show for the first time the general-applicable, synergistic design approach with an enzyme-matched carrier and productive/oriented enzyme immobilization to create the high-performance catalytic continuous-flow reactor. Therefore, this work is of distinct novelty, and worth publication in Nature Communications, after well addressing the following minor issues.

1. It may be worthwhile to compare the block copolymer membranes with other nano-sized isoporous membranes explicitly, e.g., inorganic AAO membranes and track-etched polymeric membranes, making the work better understood to the broad readership of Nature Communications.

Authors' response: **Done.** Following the reviewer's suggestion, we made the comparison explicitly in the revised introduction part (page 4).

“In this work, BCP membranes were fabricated via the combination of evaporation induced self-assembly and non-solvent induced phase separation (SNIPS) method²⁴. This method is straight forward one-step and scalable to produce asymmetric but integral membranes, which exhibit a unique structure with an isoporous top layer and a macroporous supporting sublayer¹⁴. These features endow BCP membranes with superior mechanical robustness and scalability over the conventional AAO isoporous membranes which are thermally stable and produced in a multiple step process¹². Compared to track-etched isoporous membranes²⁵, BCP membranes possess higher porosity with a 2 order of magnitude higher pore number density and thereby superior permeability.”

2. The authors should include the comparison of the obtained space-time-yield value with the required value of commercial processes in the abstract, to highlight their application potential.

Authors' response: **Done.** Following the reviewer's comment, we modified the text in the revised abstract (page 2).

“The synergistic design of enzyme-matched carriers and efficient enzyme immobilization empowers an excellent catalytic performance with superior operational stability (> 1 month), outstanding productivity, and a high space-time yield (1.05×10^5 g L⁻¹ d⁻¹) via a single-pass continuous-flow process. The obtained performance makes the designed nano- and isoporous block copolymer membrane reactor highly attractive for industrial applications.”

3. In lines 155-157, the authors state that bound YmPh-LCI formed a homogeneous layer with a thickness of about 7.0 nm, which is in good agreement with the calculated size and the measured hydrodynamic diameter of the enzyme. It would be better to describe how the authors calculated the size of the enzyme.

Authors' response: **Description included.** First the YmPh-LCI protein structure was calculated by using AlphaFold 2 software package (<https://github.com/sokrypton/ColabFold>). Based on the obtained structure, the YmPh-LCI dimensions were determined by the PyMol software suite (Version 2.5.5) by executing the script: the Draw_Protein_Dimensions.py (https://pymolwiki.org/index.php/Draw_Protein_Dimensions).

Following the reviewer's comment, we included the calculation method in the revised supplementary information (Page 9):

“1.11. Protein structure generation and dimension calculation

YmPh-LCI protein structure was generated using the AlphaFold2 software package on google's Alphafold server (<https://github.com/sokrypton/ColabFold>) with the amino acid sequence found in Appendix 3.3. Based on the obtained structure, the protein dimensions of YmPh-LCI were determined with the PyMol software suite (Version 2.5.5) by executing the script: Draw_Protein_Dimensions.py (https://pymolwiki.org/index.php/Draw_Protein_Dimensions).”

4. In line 259, “for the nanochannel of 342 ± 23 nm with a nanochannel density of $1.44 \times 10^{14} \text{ m}^{-2}$ ”, the authors should clarify the meaning of “ 342 ± 23 nm”, which may indicate either the length or diameter of nanochannels.

Authors' response: **Done**, thank you for pointing out this potential misunderstanding. “ 342 ± 23 nm” is the length of the nanochannels. In order to make the value clear for readers, we changed the sentence in the revised manuscript (page 14):

“Therefore, for the nanochannel with a length of 342 ± 23 nm and a nanochannel density of $1.44 \times 10^{14} \text{ m}^{-2}$, an optimal concentration of 23.15 mM should be used to maximize the phosphate yields from each InsP-Intermediate.”

5. In line 275, the authors state the excellent binding stability under high pressure-driven flow. it would be better to state it more directly and explicitly.

Authors' response: **Addressed.**

Following the reviewer's comment, we modified the text in the revised manuscript (page 15).

“The catalytic performance is almost completely recovered after a high pressure-driven flow test of $215 \text{ L m}^{-2} \text{ h}^{-1}$ (star point at Fig. 3b). The latter indicates neglectable phytase detachment under high pressure-driven flow, demonstrating the excellent binding strength of YmPh-LCI@M.”

6. In line 295, 300, and 303, the context of “Error! Reference source not found” should be corrected.

Authors' response: **Corrected.** We thank the reviewer to point this out. See (Page 16):

“To further demonstrate the potential of isoporous BCP membranes as ideal carriers, we tailored the nanochannel properties – size and surface charge (Fig. 4a) and further investigated their influence on the catalytic performance of NaMeR (e.g., $C_{\text{PO}_4^{3-}}$, R_{P} , productivity, and long-term operational stability). Specifically, the nanochannel size was reduced from a diameter of 57 nm to 30 nm by designing the molecular weight and composition of the PS-*b*-P4VP block copolymers, corresponding to the membranes Pri57, Pri44, Pri30 (Fig. 4c-e, Supplementary Table S2, Fig. S25). Various percentages of positive charges from 10% to 91% were introduced along the nanochannel of the membrane Pri57 via regulating the quaternization of the P4VP pore-forming block, resulting in a series of membranes PC10, PC39, PC73, and PC91 (Fig. 4a, b, f, Supplementary Fig. S21-25)⁴⁹.”

REVIEWER COMMENTS

Reviewer #1 (Remarks to the Author):

The authors have extensively addressed all the comments and revised the manuscript accordingly. The improved manuscript is now suitable for publication in Nature Communications.

Reviewer #2 (Remarks to the Author):

In this manuscript, the authors reported the fabrication of a high-performance enzymatic continuous-flow reactor that integrating enzymes on inner surface of isoporous block copolymer (BCP) membranes by using the material binding peptides as connectors. This work is interesting. The revised version of the manuscript has addressed most of the concerns raised by the reviewers. Before publishing in Nature Communications, the authors are encouraged to address following concerns:

1. As described "The diameter is only ~9-times the size of the YmPh-LCI which provides a nanoconfined environment for the enzymatic phytate hydrolysis reaction ", is it reasonable to use the nanoconfined effect to explain this phenomenon when the size gap between the enzyme and the pore is so large?
2. I notice that the change in activity caused by changing the diameter of the pore is not so large, is it reasonable?
3. The improvement of activity from carriers with different positive charged nanochannel surfaces is exciting, but the corresponding explanation seems inadequate.
4. page 11, Fig 3e: It is not clear what is exactly shown in the figure. The author should clearly indicate which storage stability tests were done at room temperature or 45 oC.
5. The author noted that "Significantly improved catalytic reactivity of YmPh-LCI over YmPh-WT can directly be attributed to its efficient immobilization through the adhesion promoting LCI peptide" in this paper. Please give the immobilization efficiency data. And the difference of reactivity between free YmPh-LCI and YmPh-LCI@M is encouraged to be added.
6. In this paper, it is described that "A general-applicable, scalable, and robust flow reactor design concept that enables the fabrication of a high-performance, continuous-flow, and

biofunctionalized NaMeR was developed and successfully validated by achieving a productivity of about 2147 mol phosphate per m² membrane within 30 days and a space-time yield of up to 1.05×10^5 g L⁻¹ d⁻¹ over 30 days.” To verify the versatility of this flow reactor, has the authors tried to use NaMeR for immobilization of other enzymes?

Rebuttal letter

We thank the reviewers for their positive, helpful, and thoughtful comments. We have responded to all the queries and comments in detail.

Reviewer #2 (Remarks to the Author):

In this manuscript, the authors reported the fabrication of a high-performance enzymatic continuous-flow reactor that integrating enzymes on inner surface of isoporous block copolymer (BCP) membranes by using the material binding peptides as connectors. This work is interesting. The revised version of the manuscript has addressed most of the concerns raised by the reviewers. Before publishing in Nature Communications, the authors are encouraged to address following concerns:

1. As described "The diameter is only ~9-times the size of the YmPh-LCI which provides a nanoconfined environment for the enzymatic phytate hydrolysis reaction ", is it reasonable to use the nanoconfined effect to explain this phenomenon when the size gap between the enzyme and the pore is so large?

Authors' response: **Clarified.** It is from the authors' points of view reasonable to use the nanoconfined effect. **Reasoning:** The term "nanoconfined effect" was used in previous reports when the enzymatic reactions take place in a confined and crowded space with dimensions less than 100 nm, which often occurs in the organelles or living cells (*ChemCatChem* 2019, 11, 5662–5670).

Further publications in which nanoconfined effects are reported include:

(a) "*The power of electrified nanoconfinement for energising, controlling and observing long enzyme cascades*, *Nat. Commun.* 2021, 12:340", in which the authors stated "five

enzymes (i.e., ferredoxin NADP⁺ reductase, L-malate-NADP⁺ oxidoreductase, L-aspartate ammonia-lyase, fumarase, carbonic anhydrase) trapped and crowded within the nanoconfined environment of a porous conducting metal oxide electrode material comprising pores and cavities of less than 100 nm in diameter”;

(b) “*Electrified Nanoconfined Biocatalysis with Rapid Cofactor*, *ChemCatChem* 2019, 11, 5662–5670”, in which the authors stated that the enzymes (flavoenzyme and ferredoxin NADP⁺ reductase (FNR)) were nanoconfined within the pores with pore dimensions in the region of 50 nm;

(c) “*A Nanoconfined Four-Enzyme Cascade Simultaneously Driven by Electrical and Chemical Energy, with Built-in Rapid, Confocal Recycling of NADP(H) and ATP*, *ACS Catal.* 2022, 12, 8811–8821”, in which the authors state that the nanoconfinement effect was involved when four enzymes (carboxylic acid reductase with MW 128 kDa, ferredoxin NADP⁺ reductase with MW 32 kDa, adenylate kinase with MW 21 kDa, and pyruvate kinase with MW 230 kDa) were loaded into and trapped in the random nanopores of an indium tin oxide (ITO) with pore diameter less than 100 nm.

(d) “*Effect of nanoconfinement on the enzymatic activity of bioactive layer-by-layer assemblies in nanopores*, *Colloids and Surf. A Physicochem. Eng. Asp.* 2022, 647 129059”, in which the authors used the nanopores with nominal diameters of 150, 250, 400, and 800 nm to study the effects of nanoconfinement on the enzymatic activity of immobilized glucose oxidase.

In summary, the term “nanoconfined effect” is commonly used as done in the same context of our manuscript.

2. I notice that the change in activity caused by changing the diameter of the pore is not so large, is it reasonable?

Authors’ response: **Clarified.** It is from our point of view reasonable that the change in activity caused by changing the diameter of the pore is not so large, since the catalytic performance of the membrane reactor is not only correlated to the pore diameter but also the length of the pores i.e. the *thickness of isoporous, cylindrical layer of the membranes*.

Reasoning in detail: Compared to Pri57 with a pore diameter of approx. 57 nm and a thickness of isoporous, cylindrical layer of approx. 350 nm (Figure 2b in the revised main manuscript), the pore diameter of Pri30 is decreased to 30 nm and the thickness of the isoporous, cylindrical layer is also decreased to approx. 200 nm. While the smaller diameter is beneficial to the catalytic performance, the smaller thickness of the isoporous layer is not. The resulting trade-off effect leads to a non-dramatic improvement of catalytic performance within a reasonable range.

To clarify this point, we have included the cross-sectional image of Pri30 in Figure S29 and the corresponding discussion in the revised supplementary information (page 43).

“

Figure S29. SEM image of cross-section of membrane Pri30 illustrating the thickness of isoporous, cylindrical layer.

Figure S29 illustrates that the membrane Pri30 has an isoporous, cylindrical layer of approx. 200 nm, which is shorter than that of Pri57 (approx. 350 nm). Based on the calculation method described in Section 2.5.3, the residence time inside the nanochannels of the cylindrical top layer of Pri30 is shortened to approx. 58% of that of Pri57; however, the radial molecular diffusion time inside the nanochannels of Pri30 is also shortened to approx. 28% of that of Pri57. The reduced radial

molecular diffusion time is beneficial to the catalytic performance while the reduced residence time is detrimental to the catalytic performance. Overall, the experimental results prove that in the case of Pri30 a slightly improved productivity compared to Pri57 can be achieved as trade-off between these two factors influencing the catalytic performance.”

3. The improvement of activity from carriers with different positive charged nanochannel surfaces is exciting, but the corresponding explanation seems inadequate.

Authors' response: **Addressed. We disagree with the comment that our explanation is inadequate, but rephrased it since we cannot measure the contact time.** From our investigations and experimental findings, the improvement of activity of immobilized enzymes onto the positively charged membrane arises not only from the effect of one single factor but rather the overall effect of three factors, i.e., prolonged contact time derived from electrostatic interaction between the oppositely charged nanochannel wall and substrate, more immobilized enzyme in the cylindrical top layer, and nanoconfinement effect, as included in the revised manuscript (page 19).

Specifically, there is a thorough and in-depth investigation (*J. Am. Chem. Soc.* 128, 14612-14618 (2006)), which designed the carrier surface and substrate with tunable charged characteristics and demonstrated that the electrostatic interaction between charged surface and substrate controlled the mass transfer of the substrate and product, and thereby plays a significant role in the catalytic performance of the immobilized enzyme. Similarly in our study, there is an electrostatic attraction between oppositely charged nanochannel surfaces and substrates, therefore, referring to the above-mentioned study and the reported role of electrostatics led us to give the conclusion that electrostatic attraction might affect interactions of substrates and nanochannel walls, resulting in prolonged contact time. However, we agree that the contact time cannot be experimentally measured or determined but can be supported by the above-mentioned thorough study.

For the factors of more immobilized enzyme in the cylindrical top layer and nanoconfinement effect, our experimental findings in terms of the TEM images in Fig. 5f and Fig. 2g and SPR result in Fig. 5g of the revised manuscript proved and explained the more immobilized enzyme in the cylindrical top layer of the positively charged membranes. Accordingly, the nanoconfinement effect from the cylindrical top layer is amplified due to the increased amount of enzyme.

To clarify these factors, we modified the text in the revised manuscript (page 19).

It is stated:

“Oppositely charged surface/substrate enhancing the catalytic performance of the immobilized enzyme onto a solid surface was reported, attributing to the electrostatic interaction-enhanced mass transfer in terms of substrate capture and product release⁵⁰. In our study, given that the electrostatic attractions between the oppositely charged nanochannel wall and substrate (i.e., the positively charged nanochannel surface and negatively charged InsP6/InsP intermediates), we expect that the corresponding electrostatic attractions might prolong the contact time of substrates with the nanochannel wall, thus enhancing hydrolysis of InsP6 and InsP-intermediates. Moreover, InsP6 with its six negatively charged phosphate groups might experience a longer averaged contact time with the positively charged nanochannel walls than InsP intermediates with fewer phosphate groups due to a stronger electrostatic attraction between InsP6 and nanochannels. A longer contact time and stepwise hydrolysis of phosphate from InsP6 hydrolysis might result in a steep and a stepwise local concentration gradient from the uppermost segment of the nanochannel. As a result, a high specific phytase activity was ensured for a rapid and efficient stepwise hydrolysis of InsP6 to InsP2.”

4. page 11, Fig 3e: It is not clear what is exactly shown in the figure. The author should clearly indicate which storage stability tests were done at room temperature or 45 °C.

Authors' response: **Clarified**. We previously performed the storage stability tests at both room temperature and 4 °C for 45 days and included the data; see modified Fig. 3e in the revised manuscript (page 11) and Figure S18 of the revised supplementary information (page 34).

From the original Fig. 3e (page 11):

To the modified Fig. 3e (page 11):

From the original Figure S18 (supplementary information, page 34):

To the modified Figure S18 (supplementary information, page 34):

5. The author noted that "Significantly improved catalytic reactivity of YmPh-LCI over YmPh-WT can directly be attributed to its efficient immobilization through the adhesion promoting LCI peptide" in this paper. Please give the immobilization efficiency data. And the difference of reactivity between free YmPh-LCI and YmPh-LCI@M is encouraged to be added.

Authors' response: **Data included.** The immobilization efficiency and difference of reactivity between free YmPh-LCI and YmPh-LCI@M are included in Table S3 (page 19)

and Table S4 (page 19) of the revised supplementary information, respectively. Please note that as expected the calculated reactivity of YmPh-LCI@M is lower than that of free YmPh-LCI. The data of the immobilized YmPh-LCI@M does not reflect its performance since the flow through the membrane is restricted to diffusion. It can be regarded as a quick experiment to see whether the immobilized enzymes are active under substrate depletion and product accumulation conditions. A fair performance comparison is the flow reactor comparison to the free enzyme as shown in Fig. S17 and Table S10 of the revised supplementary information (page 32-33).

In the revised Supplementary Information (page 19, Table S3 and Table S4):

“Table S3. Binding capacity and binding efficiency of YmPh-WT and YmPh-LCI onto isoporous BCP membranes and MTP plates made of polypropylene.

Entry	YmPh-WT		YmPh-LCI	
	BCP membrane	MTP	BCP membrane	MTP
1 C_0 (g L ⁻¹)	0.271 ± 0.005	0.271 ± 0.005	0.298 ± 0.004	0.298 ± 0.004
2 C_1 (g L ⁻¹)	0.228 ± 0.006	0.267 ± 0.009	0.155 ± 0.010	0.290 ± 0.006
3 C_{w1} (g L ⁻¹)	0.043 ± 0.010	0.008 ± 0.007	0.007 ± 0.005	0.006 ± 0.005
4 C_{w2} (g L ⁻¹)	0.017 ± 0.007	0.004 ± 0.003	0.001 ± 0.001	0.003 ± 0.002
5 C_{w3} (g L ⁻¹)	0.005 ± 0.007	0.002 ± 0.001	0.007 ± 0.004	0.002 ± 0.001
Binding capacity (pmol cm ⁻²)	< 10	< 10	830	< 10
Binding efficiency	< 0.5%	< 0.5%	42%	< 0.5%

YmPh-WT and YmPh-LCI binding onto MTP plates (polypropylene) were negative controls to reflect the background binding of proteins on the surface of MTP wells¹³, while isoporous BCP membranes were placed inside MTP wells for binding detections. The amount of YmPh-LCI binding to MTP (polypropylene) or isoporous BCP membrane (diameter 6 mm) using shaking incubation process was quantified at 280 nm using a spectrophotometer (NanodropTM 2000; ThermoFisher Scientific) and purified proteins (YmPh-WT: 47.3 kDa, extinction coefficient 49890 M⁻¹ cm⁻¹; YmPh-LCI: 55.1 kDa, extinction coefficient 73840 M⁻¹ cm⁻¹).

Table S4. Quantification of reactivity of free YmPh-LCI and YmPh-LCI@M in Pri57 membrane.

Entry		Reactivity (RFU s ⁻¹ pmol _{enzyme} ⁻¹)
1	Free YmPh-LCI	990
2	YmPh-LCI@M	0.031

The reactivity of free YmPh-LCI was calculated from the reaction rate of 8.49×10^4 RFU s⁻¹ with 0.858 μM enzyme (100 μl, Figure S3); the reactivity of YmPh-LCI@M of Pri57 was calculated from the reaction rate of 6.98 RFU s⁻¹ with 830 pmol cm⁻² immobilized enzyme (0.28 cm² membrane, Figure S5b, Table S3).’’

Accordingly, we modified the Methods in the revised manuscript (page 24-25):

‘‘The amount of immobilized phytase enzyme (pmol cm⁻²) was determined in a stepwise process (see Supplementary Table S3) by determining the starting protein concentration (C_0 , g L⁻¹), supernatant concentration before washing (C_1 , g L⁻¹), after first washing (C_{w1} , g L⁻¹), after second washing (C_{w2} , g L⁻¹), and after third washing (C_{w3} , g L⁻¹). Protein concentrations were determined at a wavelength of 280 nm using a UV spectrophotometer (NanodropTM 2000; Thermo Fisher Scientific) for YmPh-WT (47.3 kDa, 49890 M⁻¹ cm⁻¹) and YmPh-LCI (55.1 kDa, 73840 M⁻¹ cm⁻¹). MTP wells without membrane inside were applied as negative controls. All experiments were performed in triplicate. The amount of the immobilized enzyme on the membrane was calculated using the following equation:

$$\text{Immobilized enzyme} = \frac{\text{mol (enzyme)}}{\text{membrane area}} = \frac{(C_0 - C_1 - C_{w1} - C_{w2} - C_{w3}) \times V \times 10^3}{A \times M}$$

where V is the volume of solutions (μL), M is the protein molecular weight (kDa), and A is the top surface area of the membrane (cm²).

The binding efficiency was calculated using the following equation:

$$\text{Binding efficiency} = \frac{\text{mol (binding enzyme)}}{\text{mol (feeding enzyme)}} = \frac{(C_0 - C_1 - C_{w1} - C_{w2} - C_{w3}) \times V \times M}{C_0 \times V \times M}$$

where V is the volume of solutions (μL), M is the protein molecular weight (kDa).

For determination of the enzymatic activity after immobilization on the catalytic membrane, 100 μL reaction solution containing 4-MUP (0.5 mM) in washing solution A was loaded to each well of an

MTP and incubated (MTP shaker, 600 rpm, 25 °C). The relative fluorescence over time (0.5, 5.5, 10.5, 15.5, 20.5, 25.5 min) was measured using a CLARIOstar® M1000 microtiter plate reader (BMG Labtech, λ_{ex} : 322 nm, λ_{em} : 464 nm; gain: 100, room temperature). The maximum reaction rate was calculated to demonstrate the enzymatic activity of the catalytic membrane. All experiments were performed in triplicate. The reactivity ($\text{RFU s}^{-1} \text{ pmol}_{\text{enzyme}}^{-1}$) of free YmPh-LCI and YmPh-LCI@M was calculated using the following equation:

$$\text{Reactivity} = \frac{\text{reaction rate (RFU s}^{-1}\text{)}}{\text{molars of enzyme (pmol)}}$$

Please note that the reactivity of YmPh-LCI@M does not reflect its performance since the flow through the membrane is restricted by diffusion. It can rather be regarded as a quick experiment to see whether the immobilized enzymes are active under substrate depletion and product accumulation conditions.”

6. In this paper, it is described that “A general-applicable, scalable, and robust flow reactor design concept that enables the fabrication of a high-performance, continuous-flow, and biofunctionalized NaMeR was developed and successfully validated by achieving a productivity of about 2147 mol phosphate per m² membrane within 30 days and a space-time yield of up to 1.05 × 10⁵ g L⁻¹ d⁻¹ over 30 days.” To verify the versatility of this flow reactor, has the authors tried to use NaMeR for immobilization of other enzymes?

Authors' response: **YES, we indeed investigated three enzymes** in initial immobilisation experiments and decided to perform the detailed performance characterization of NaMeR with the phytase enzyme due to its high thermal resistance which often correlates well to the storage and process stability. The results of the other two enzymes with initial immobilization results are included in Figure S33 (page 47) of the revised supplementary information. In addition, the corresponding experimental part is included in the revised supplementary information (page 9-13) and also referred to it in the revised manuscript (page 22).

In detail: We illustrated the active immobilizations of 3 LCI fused enzymes, i.e., YmPh-LCI, CaLB-LCI, and GalO_{XM3-5}-LCI, onto the membranes. It demonstrates the versatility of the proposed design concept.

In the revised supplementary information (page 9-13):

1.12 GalO_{XM3-5}-LCI expression and purification

Galactose oxidase mutant M3-5 (GalO_{XM3-5}) exhibits a great sustainable route to produce building blocks of biobased polymers through biocatalytic oxidation of 5-Hydroxymethylfurfural (HMF)^{8,9}. GalO_{XM3-5}-17Helix-TEV-LCI (GalO_{XM3-5}-LCI) gene construct integrated in pALXtreme-5b plasmid was generated in-house and overexpressed in *E. coli* BL21-Gold (DE3) LacI^{Q1}. Briefly, pre-cultures were inoculated from glycerol stocks with a sterile pipette tip into cultivation tubes containing LB medium (5 mL; 10 g L⁻¹ tryptone, 10 g L⁻¹ NaCl, 5 g L⁻¹ yeast extract) supplemented with ampicillin (100 µg mL⁻¹) and cultivated overnight (37 °C, 250 rpm). Main cultures were inoculated in 500 mL shaking-flasks containing TB medium (100 mL; 12 g L⁻¹ tryptone, 24 g L⁻¹ yeast extract, 5 g L⁻¹ glycerol, 2.31 g L⁻¹ KH₂PO₄, and 12.5 g L⁻¹ mM K₂HPO₄, supplemented with ampicillin (100 µg mL⁻¹) with an OD₆₀₀ of 0.05. When main cultures were grown (37 °C, 250 rpm) until an OD₆₀₀ of 0.6-0.8, protein expression was induced by adding 0.1 mM isopropyl β-D-1-thiogalactopyranoside (IPTG) with 0.4 mM CuSO₄. After protein expression (25 °C, 200 rpm, 20 h), the cell pellet was harvested by centrifuging (Sorvall, ThermoFischer Scientific, Germany, 4 °C, 4000 × g, 20 min). The collected cell pellet was suspended in 40 mL Tris-HCl buffer (50 mM, pH 8.0) and lysed by a French press. The lysates were centrifuged (4 °C, 10000 rpm, 30 min) and the supernatant was collected and filtered with a 0.45 µm filter before purification.

GalO_{XM3-5}-LCI was purified by using anion-exchange chromatography (HiTrap[®] Q HP column, Cytiva, United States; ÄKTAprime plus, Cytiva, United States). Briefly, the filtered supernatant was subsequently loaded into the anion-exchange column (bed volume 5 mL, equilibrated with Tris-HCl buffer (50 mM, pH 8.0)), and GalO_{XM3-5}-LCI was then eluted under the NaCl gradient (1 M; 0% to 100% in 100 mL) in Tris-HCl buffer (50 mM, pH 8.0). The elution samples (2 mL/tube) were analysed with SDS-PAGE. Tubes of purified enzymes were collected in dialysis bag (14 kDa cut-off) and incubated in dialysis buffer (50 mM sodium phosphate buffer (NaPi), pH 7.0) with CuSO₄

(0.4 mM) for 24 h at 4 °C. Afterwards, the collected samples were dialyzed against NaPi buffer (50 mM, pH 7.0; 400 mL) for 24 h at 4 °C and the dialysis buffer was changed 2-3 times. The concentration of desalted protein was detected at 280 nm using a UV spectrophotometer (Nanodrop™ 2000; Thermo Fisher Scientific) applying extinction coefficients (calculated with Benchling software, <https://benchling.com>, $M^{-1} \text{ cm}^{-1}$) and molecular masses (kDa) of purified proteins (GalO_{xM3-5}-LCI: 76.4 kDa, $142585 \text{ M}^{-1} \text{ cm}^{-1}$). The purified proteins were stored at -20 °C for further use.

1.13 GalO_{xM3-5}-LCI binding to isoporous membrane and the activity detection by ABTS assay

100 µL washing solution C (50 mM NaPi buffer, pH 7.0, 4 mM Triton X100) was applied to prewash the membrane with pipetting up and down (membrane pieces preparation see above). Afterwards, the prewashed membrane was washed again with 100 µL washing solution D (50 mM NaPi buffer, pH 7.0) twice. Subsequently, the membrane was washed with 100 µL binding solution (50 mM Tris-HCl buffer, pH 8.0) followed by 100 µL protein solution (ranging from 0-8.58 µM) diluted in the binding solution. After incubation on a MTP shaker (ELMI SkyLine DTS-4 Digital Thermo Shaker; Elminorthamerica Ltd.; 10 min, 600 rpm, 25 °C), the protein-loaded membrane was washed with 100 µL washing solution C (MTP shaker, 5 min, 600 rpm, 25 °C). The washing steps were repeated three times. Finally, the washed membrane was pinched from MTP using tweezers and dried using nitrogen gas. Membrane pieces were transferred to a new transparent MTP (white, PS, Greiner bio-one GmbH) facing up before starting the ABTS (2,2'-azino-bis (3-ethylbenzothiazoline-6-sulfonic acid)) assay¹⁰.

For determination of the enzymatic activity after immobilization on the membrane, 100 µL reaction solution containing ABTS (1 mM), HRP (0.005 g L⁻¹), and HMF (5 mM) in NaPi buffer (50 mM, pH 7.0) was loaded to each well of an MTP. The relative absorbance over time was measured under wavelength of 420 nm using a Tecan Sunrise microtiter plate reader (Tecan Trading AG, kinetic cycles: 400, kinetic interval: 3 s, room temperature). The initial reaction rate was calculated to demonstrate the enzymatic activity of the catalytic membrane. All experiments were performed in triplicate.

1.14 *Candida antarctica* lipase B-LCI expression and purification

Candida antarctica lipase B (CaLB) is well-known as an industrially important enzyme¹¹. CaLB-17Helix-TEV-LCI (CaLB-LCI) gene construct integrated in pGAPzαA plasmid was generated in-

house and expressed in *Pichia pastoris* SMD1168¹². *P. pastoris* cells containing CaLB-LCI plasmid were pre-grown on YPD agar plate (48 h, 30 °C; 10 g/L yeast extract, 20 g L⁻¹ peptone, 20 g L⁻¹ dextrose, 20 g L⁻¹ agar, 100 µg mL⁻¹ zeocin) and then a single colony was inoculated in YPD medium (16 h, 30 °C, 200 rpm, 10 mL; 10 g/L yeast extract, 20 g L⁻¹ peptone, 20 g L⁻¹ dextrose, 50 µg mL⁻¹ zeocin) as a pre-culture. The main culture was prepared with an inoculation 0.4 mL pre-culture in 200 mL YPD medium in flask (1 L) and incubated under 30 °C, 200 rpm for 72 h. After main culture, cells were harvested (4 °C, 4000 rpm, 30 min) and the supernatant containing secreted enzymes was collected. Tris-acetate buffer (250 mM, pH 7.0) was added in the supernatant with a volume ratio of 1:10 and then filtered with a 0.45 µm filter before purification.

The supernatant containing CaLB-LCI was concentrated (15 folds) and equilibrated slowly with ammonium acetate (0.8 M). CaLB-LCI was purified by using hydrophobic interaction chromatography (HIC) (packed volume 100 mL; Fractogel TSK Butyl 650 Size S, Merck, Darmstadt, Germany;) and ÄKTA system (GE Healthcare, Chalfont St Giles, UK). The column was equilibrated with Tris-acetate buffer (250 mL, 25 mM, pH 7.0). After loading the concentrated supernatant (flow rate 2 mL min⁻¹), the protein was eluted with a gradient elution from 100% Tris-acetate buffer (25 mM, pH 7.0) containing ammonium acetate (0.8 M) to 100% water (0%-100% in 100 mL) and collected in aliquots of 2 mL/tube. After SDS-APGE analysis of elution aliquots, purified CaLB-LCI samples were collected and buffer exchanged with triethanolamine buffer (TEA; 100 mM, pH 7.5). The concentration of purified protein was detected at 280 nm using a UV spectrophotometer (NanodropTM 2000; Thermo Fisher Scientific) applying extinction coefficients (calculated with Benchling software, <https://benchling.com>, M⁻¹ cm⁻¹) and molecular masses (kDa) of purified proteins (CaLB-LCI: 40.9 kDa, 65235 M⁻¹ cm⁻¹). The purified proteins were stored at -20 °C for further use.

1.15 CaLB-LCI binding to isoporous membrane and the activity detection by pNPB assay

100 µL washing solution E (100 mM TEA buffer, pH 7.5, 0.1% (v/v) Tween 20) was applied to prewash the membrane with pipetting up and down (membrane pieces preparation see above). Afterwards, the prewashed membrane was washed again with 100 µL washing solution F (100 mM TEA buffer, pH 7.5) twice. Subsequently, the membrane was washed with 100 µL binding solution (50 mM Tris-HCl buffer, pH 8.0) followed by 100 µL protein solution (ranging from 0-21.45 µM) diluted in the binding solution. After incubation on a MTP shaker (ELMI SkyLine DTS-4 Digital

Thermo Shaker; Elminorthamerica Ltd.; 10 min, 600 rpm, 25 °C), the protein-loaded membrane was washed with 100 μ L washing solution E (MTP shaker, 5 min, 600 rpm, 25 °C). The washing steps were repeated three times. Finally, the washed membrane was pinched from MTP using tweezers and dried using nitrogen gas. Membrane pieces were transferred to a new transparent MTP (white, PS, Greiner bio-one GmbH) facing up before starting the *p*NPB (*p*-Nitrophenyl Butyrate) assay.

For determination of the enzymatic activity after immobilization on the membrane, 100 μ L reaction solution containing *p*NPB (0.5 mM, originally dissolved in acetonitrile) in TEA buffer (100 mM, pH 7.5) was loaded to each well of an MTP. The relative absorbance over time was measured under wavelength of 410 nm using a Tecan Sunrise microtiter plate reader (Tecan Trading AG, kinetic cycles: 100, kinetic interval: 12 s, room temperature). The initial reaction rate was calculated to demonstrate the enzymatic activity of the catalytic membrane. All experiments were performed in triplicate.”

In the revised supplementary information (page 47, Figure S33):

“

Figure S33. Dose-response curve to analyze binding efficiency of GalO_xM₃₋₅-LCI and CaLB-LCI on isoporous BCP membranes (Pri57). Error bars represent s.d. of the mean.”

In the revised manuscript (page 22):

“In summary, we believe that the designed concept is general-applicable and scalable due to the design flexibility of the isoporous BCP membranes, and scalable/oriented immobilizations of enzymes through MBP (Supplementary Fig. S33).”

REVIEWERS' COMMENTS

Reviewer #2 (Remarks to the Author):

After carefully comparing the current manuscript and the related replies with the previous submission, the reviewer is glad to see the improvements made by the authors, and is willing to recommend the manuscript to be published in its current form.